# Score Propagation as a Catalyst for Graph Out-of-distribution Detection: A Theoretical and Empirical Study

## Abstract

The field of graph learning has been substantially advanced by the development of deep learning models, in particular graph neural networks. However, one salient yet largely under-explored challenge is detecting Out-of-Distribution (OOD) nodes on graphs. Prevailing OOD detection techniques developed in other domains like computer vision, do not cater to the interconnected nature of graphs. This work aims to fill this gap by exploring the potential of a simple yet effective method – OOD score propagation, which propagates OOD scores among neighboring nodes along the graph structure. This post hoc solution can be easily integrated with existing OOD scoring functions, showcasing its excellent flexibility and effectiveness in most scenarios. However, the conditions under which score propagation proves beneficial remain not fully elucidated. Our study meticulously derives these conditions and, inspired by this discovery, introduces an innovative edge augmentation strategy with theoretical guarantee. Empirical evaluations affirm the superiority of our proposed method, outperforming strong OOD detection baselines in various scenarios and settings.

## 1 Introduction

Graph-like data structures are ubiquitous in many domains, such as social networks (Zafarani et al., 2014; Li & Goldwasser, 2019), molecular chemistry (Gasteiger et al., 2019a; Yan et al., 2019), and recommendation systems (Ying et al., 2018; Liu et al., 2021b). As graph neural networks increasingly serve as powerful tools for navigating this complex data landscape, a compelling yet under-explored issue emerges: Out-of-Distribution (OOD) node detection. Imagine a recommender system suggesting irrelevant or even harmful products to users, or a bioinformatics algorithm misusing an unknown protein. This gives rise to the importance of OOD detection in graph data, which determines whether an input is in-distribution (ID) or OOD and enables the model to take precautions.

While existing OOD detection methods have shown promising results in computer vision (Sun et al., 2022b; Jaeger et al., 2022; Galil et al., 2022; Djurisic et al., 2022; Zhu et al., 2022), natural language procession (Colombo et al., 2022; Ren et al., 2022) and tabular data analytics (Ulmer et al., 2020), their effectiveness diminishes when applied to graph data (Wu et al., 2022). These conventional techniques operate under the assumption that data points are independently sampled, which misaligns with the interconnected nature of graphs. For example, in a social network, nodes (people) do not exist in isolation but are linked through friendships, interests, etc.

Applying traditional OOD detection methods such as KNN (Sun et al., 2022b) or Mahalanobis (Lee et al., 2018; Sehwag et al., 2021; Ren et al., 2021) distances to the learned node embeddings without fully considering the node dependencies can be inadequate.

To leverage abundant structural knowledge in graph data, we investigate one straightforward method for graph OOD detection through *OOD score propagation* – aggregating the OOD

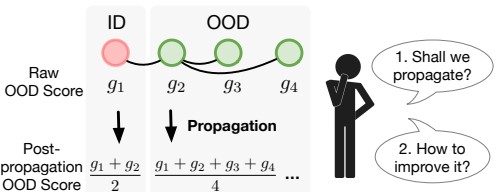

Figure 1: Illustration of the propagation procedure for OOD scores and two questions to be answered.

scores from connected neighbor nodes (as shown in Figure 1). This strategy offers several notable benefits: (a) it can seamlessly integrate with all existing OOD scoring functions, ensuring compatibility and flexibility across a wide array of use cases (Zhu et al., 2003); (b) it obviates the need for cumbersome retraining procedures, offering a flexible *post hoc* approach to OOD detection. In light of its potential, our research embarks on addressing two pivotal research questions related to OOD score propagation:

- *Question 1: "Will naive OOD score propagation always help graph OOD Detection?"* Our investigation, as detailed in Section 3, provides theoretical insights into this query. We reveal the essential condition for propagation to be beneficial: the ratio of intra-edges (`ID-to-ID` and `OOD-to-OOD`) must surpass that of inter-edges (`ID-to-OOD`). This finding naturally paves the way for our subsequent inquiry.

- *Question 2: "How to augment the propagation strategy for better graph OOD detection?"* Building on our prior findings, we propose a graph augmentation strategy as presented in Section 4. Specifically, our strategy selects a subset $G$ of the training set and puts additional edges to the nodes within $G$. Beyond its practical implications, our solution is also theoretically supported: When $G$ predominantly connects to ID data over OOD data, our strategy can provably enhance the post-propagation OOD detection outcomes.

We summarize our contributions as below:

- **Theoretical understanding:** We delve deeply into the mechanism of score propagation to understand its potential for graph OOD detection. Our research not only validates the efficacy of this approach but also elucidates the conditions under which it thrives, providing an understanding that extends beyond existing knowledge.

- **Practical solution:** To counter the identified challenge of improving post-propagation OOD detection performance, we propose **GR**aph-**A**ugmented **S**core **P**ropagation (GRASP), an innovative edge augmentation strategy with theoretical guarantee. By strategically adding edges to a chosen subset $G$ of the training set, as detailed in Section 4, our method aims to enhance the intra-edge ratio, thereby boosting OOD detection outcomes post-propagation.

- **Empirical studies**: We demonstrate the superior performance of the proposed method on extensive graph OOD detection benchmarks, different pre-trained methodologies (Kipf & Welling, 2017; Zhu et al., 2020), and different OOD scoring functions. Under the same condition, our proposed strategy substantially reduces the FPR95 by **8.43**% compared to the strongest graph OOD detection baselines. Extensive ablation studies are also provided to show the superiority of the proposed methodology designs and the validity of the theoretical findings.

## 2 PRELIMINARIES

**Problem setup**. We consider a traditional semi-supervised node classification setting with the additional unlabeled nodes from the out-of-distribution class. Let $\mathcal{G} = \{\mathcal{V}, \mathcal{E}\}$ denote the graph with nodes $\mathcal{V}$ and edges $\mathcal{E}$, where the node set $\mathcal{V}$ with size $N$ are attributed with data matrix $X \in \mathbb{R}^{N \times d}$. The structure of graph $\mathcal{G}$ is described by the adjacency matrix $A \in \{0, 1\}^{N \times N}$. We let the corresponding row-stochastic matrices as $\bar{A} = \mathrm{D}^{-1} A$, where $D$ is the diagonal matrix with $D_{ii} = \sum_j A_{ij}$. The $N$ nodes are partially labeled, so we let $\mathcal{V}_l$ and $\mathcal{V}_u$ represent the labeled and unlabeled node sets respectively, i.e, $\mathcal{V} = \mathcal{V}_l \cup \mathcal{V}_u$. Given a training set $\mathbb{D}_{tr} = \{(\mathbf{x}_i, y_i)\}_{i \in \mathcal{V}_l}$ with $\mathbf{x}_i$ as the $i$-th row of $X$ and $y_i \in \mathcal{Y} \triangleq \{1, \cdots, C\}$, the goal of node classification is to learn a mapping $f : \mathcal{V} \to \mathbb{R}^C$ from the nodes to the probability of each class.

**Out-of-distribution detection**. When deploying a model in the real world, a reliable classifier should not only accurately classify known in-distribution (ID) nodes, but also identify "unknown" nodes or OOD nodes. Formally, we can represent the unlabeled node set by $\mathcal{V}_u = \mathcal{V}_{uid} \cup \mathcal{V}_{uood}$ where $\mathcal{V}_{uid}$ and $\mathcal{V}_{uood}$ represent the in-distribution (ID) node and OOD node respectively. **The goal of the graph OOD detection** is to derive an algorithm to decide if a node $i \in \mathcal{V}_u$ is from $\mathcal{V}_{uood}$ or $\mathcal{V}_{uid}$.

This can be achieved by having an OOD detector, in tandem with the node classification model $f$. OOD detection can be formulated as a binary classification problem. At test time, the goal of OOD

detection is to decide whether an unlabeled node $i \in \mathcal{V}_u$ is from ID or OOD. The decision can be made via a level set estimation:

$$F_{OODD}(i, \mathcal{G}; \lambda) = \begin{cases} \text{ID} & g(\mathbf{x}_i) \geq \lambda \\ \text{OOD} & g(\mathbf{x}_i) < \lambda \end{cases},$$

where nodes with higher scores $g(\mathbf{x}_i)$ are classified as ID and vice versa, and $\lambda$ is the threshold commonly chosen so that a high fraction (e.g., 95%) of ID data is correctly classified.

In this paper, we consider **post hoc** OOD detection methods to produce $g(\mathbf{x}_i)$ which does not require expensive re-training. As an example, a classical way to compute $g(\mathbf{x}_i)$ is Maximum Softmax Probability (MSP) (Hendrycks & Gimpel, 2016) which is given by the maximum softmax value. We include details of the considered OOD detection methods in Appendix A.

## 3   WILL PROPAGATION ALWAYS HELP GRAPH OOD DETECTION?

The majority of techniques for Out-of-Distribution (OOD) detection are primarily tailored for images (Hendrycks & Gimpel, 2016; Sun et al., 2022a; Zhu et al., 2022) and tabular data analytics (Ulmer et al., 2020). While it is certainly possible to adapt these methodologies to the graph data, they are not inherently designed to capture the node dependencies, a key that could potentially boost the effectiveness of graph OOD detection.

Previous empirical studies have demonstrated that improvements in graph OOD Detection could be realized through the propagation of the OOD scoring vector (Wu et al., 2022) along the graph structure. However, rather than merely corroborating these preliminary findings, our research delves into a deeper understanding of the underlying mechanisms. Specifically, we aim to answer the following research question: *Will propagation always help graph OOD Detection?* We start by showing the formal definition of propagation.

**Define OOD scoring propagation**. Given a raw OOD scoring vector $\hat{\mathbf{g}} \in \mathbb{R}^N$ with $\hat{\mathbf{g}}_i = g(\mathbf{x}_i)$, the propagated scoring vector is given by:

$$\text{Propagated OOD Scoring Vector: } \mathbf{g} = \bar{A}^k \hat{\mathbf{g}}, \tag{1}$$

where $k \in \mathbb{N}^+$ are hyperparameters.

*Is it necessarily the case that $\mathbf{g}$ outperforms $\hat{\mathbf{g}}$?* The answer is **NO**. We elucidate with the theoretical insight below.

**Theoretical Insight.** To elucidate this, we refer to a toy example illustrated in Figure 2. The discussion is decomposed into two distinct scenarios: (a) In Figure 2(a) where the number of `ID-to-ID` and `OOD-to-OOD` edges surpasses that of `ID-to-OOD` edges, the propagation mechanism tends to "aggregate" the scores associated with the ID data which further amplify the separability between the ID and OOD nodes. (b) Conversely, when the number of `ID-to-OOD` edges are more than the other types of edges, the scores for both ID and OOD nodes become undistinguishable post-propagation.

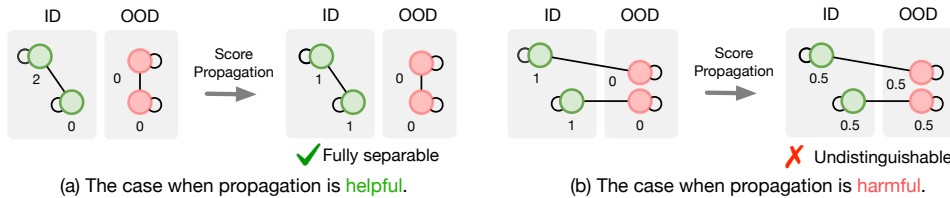

Figure 2: Two illustrative examples when scoring propagation is helpful/harmful. We consider two ID nodes in green and two OOD nodes in red. The value represents the respective OOD scores. Consequently, the propagated scores in these cases will be the mean of the scores of adjacent nodes.

The example above offers the insight that the relative performance of $\mathbf{g}$ compared to $\hat{\mathbf{g}}$ is contingent upon the structural dynamics of the network, specifically the distribution of edges. To formally articulate this relationship, we adopt a probabilistic framework for modeling edges. Specifically, we

assume that the edge follows a Bernoulli distribution characterized by parameters $\eta_{intra}$ and $\eta_{inter}$ for intra-edges (ID-to-ID and OOD-to-OOD) and inter-edges (ID-to-OOD), respectively:

$$A_{ij} \sim \begin{cases} Ber(\eta_{intra}), & \text{if } i,j \in \mathcal{V}_{uid} \text{ or } i,j \in \mathcal{V}_{uood} \\ Ber(\eta_{inter}), & \text{if } i \in \mathcal{V}_{uid}, j \in \mathcal{V}_{uood} \text{ or } j \in \mathcal{V}_{uid}, i \in \mathcal{V}_{uood} \end{cases}$$

In the context of probabilistic modeling, the subsequent Theorem 3.1 can be established to formalize the inherent understanding.

> **Theorem 3.1.** *(Informal) (a) When $\eta_{intra} \gg \eta_{inter}$, it is highly likely that the propagation algorithm will yield enhanced performance in OOD detection. (b) When $\eta_{intra} \approx \eta_{inter}$ or even $\eta_{intra} < \eta_{inter}$, the score propagation is likely to be either ineffective or detrimental to the performance.*

We also provide the formal version below (Theorem 3.2) which provides a mathematical foundation for understanding how varying the Bernoulli parameters influence the efficacy of the propagation in the context of OOD detection. We provide the detailed proof in Appendix B.

> **Theorem 3.2.** *(Formal) For any two test ID/OOD node set $S_{id} \subset \mathcal{V}_{uid}, S_{ood} \subset \mathcal{V}_{uood}$ with equal size $N_s$, let the ID-vs-OOD separability $\mathcal{M}_{sep}$ defined on an OOD scoring vector $\hat{\mathbf{g}} \in \mathbb{R}^N$ as*
>
> $$\mathcal{M}_{sep}(\hat{\mathbf{g}}) \triangleq \mathbb{E}_{i \in S_{id}} \hat{\mathbf{g}}_i - \mathbb{E}_{j \in S_{ood}} \hat{\mathbf{g}}_j.$$
>
> *If $\mathcal{M}_{sep}(\hat{\mathbf{g}}) > 0$ and $\eta_{intra} - \eta_{inter} > 1/N_s$, for some $\epsilon > 0$ and constant $c$, we have*
> $$\mathbb{P}\left(\mathcal{M}_{sep}(A\hat{\mathbf{g}}) \geq \mathcal{M}_{sep}(\hat{\mathbf{g}}) - \epsilon\right) \geq 1 - exp(-\frac{c\epsilon^2}{\|\hat{\mathbf{g}}\|_2^2}).$$

**Summary.** This section has presented a comprehensive overview of both empirical and theoretical evidence to substantiate the claim that propagation through the adjacency matrix $A$ does not necessarily enhance out-of-distribution (OOD) detection in graphs. Moreover, Theorem 3.2 reveals that the critical factor in enhancing post-propagation performance lies in **improving the ratio of intra-edges to inter-edges** within the graph structure. These insights serve as a direct motivation for the augmentation strategy that will be proposed in the next section.

## 4 HOW TO BOOST POST-PROPAGATION OOD DETECTION PERFORMANCE?

The findings from the preceding section give rise to a subsequent thought: "*Can we improve the propagation strategy for graph OOD detection performance?*" In an ideal scenario, if an oracle were to indicate that a particular subset in the test set belongs exclusively to the ID or OOD, one could augment the graph by adding intra-edges or removing inter-edges. This would consequently improve the ratio of $\eta_{intra}/\eta_{inter}$, leading to enhanced OOD detection performance post-propagation.

However, such an oracle does not exist in practical settings, and even approximating such a subset proves to be a difficult task. Existing literature has suggested the use of pseudo-labels assigned to nodes (Lee et al., 2013; Xie et al., 2020; Arazo et al., 2020; Wang et al., 2021a; Pham et al., 2021). Nonetheless, these studies also caution that this approach is susceptible to "confirmation bias", whereby errors in estimation are inadvertently amplified.

To circumvent it, this paper proposes the solution for adding edges to a subset of the training set $\mathcal{V}_l$, which is assured to be in-distribution data. We start by showing the theoretical underpinnings that adding such a subset can, under specified conditions, contribute to improved OOD detection performance after propagation.

### 4.1 THEORETICAL INSIGHT

Our approach involves adding the edges to a subset $G$ of training data and then propagating the out-of-distribution (OOD) scoring vector using the enhanced

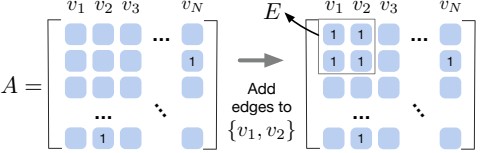

Figure 3: The augmentation procedure.

adjacency matrix. Specifically, when edges are added to $G$, this action can be mathematically represented as incorporating a perturbation matrix $E = \mathbf{e}_G \mathbf{e}_G^\top$ into A, as demonstrated in Figure 3. Here, $\mathbf{e}_S \in \mathbb{R}^N$ denotes an indicator vector for a set $S \subset \mathcal{V}$, where the vector takes the value of 1 if the index $i \in S$ and value 0 otherwise. A sufficient condition for the efficacy of this augmentation strategy in enhancing post-propagation OOD detection performance is outlined in Theorem 4.1.

> **Theorem 4.1.** *(Informal) For a subset $G$ in the training set, augmenting $G$ by adding edges to all its nodes can lead to improved post-propagation OOD detection performance, provided that the following condition is met: $\underline{G}$ has more edges to ID data than OOD data.*

We also provide the formal version below (Theorem 4.2) that incorporates a perturbation analysis. This analysis elucidates how edge augmentation in the training set can positively influence the propagation algorithm's ability to enhance OOD detection. For the sake of the main intuition, we provide the analysis on $A$ instead of $\bar{A}$ for simplicity. We provide the detailed proof in Appendix B.

> **Theorem 4.2.** *(Formal) For any two test ID/OOD node set $S_{id} \subset \mathcal{V}_{uid}, S_{ood} \subset \mathcal{V}_{uood}$ with size $N_s$, let the ID-vs-OOD separability $\mathcal{M}_{sep}$ defined on a non-negative OOD scoring vector $\hat{\mathbf{g}} \in \mathbb{R}^N$ as*
>
> $$\mathcal{M}_{sep}(\hat{\mathbf{g}}) \triangleq \mathbb{E}_{i \in S_{id}} \hat{\mathbf{g}}_i - \mathbb{E}_{j \in S_{ood}} \hat{\mathbf{g}}_j.$$
>
> *Let $\mathcal{E}_{S \leftrightarrow S'} \subset \mathcal{E}$ to denote the edge set of edges between two node sets $S$ and $S'$, where $S, S' \subset \mathcal{V}$. If we can find a node set $G \subset \mathcal{V}_l$ such that $|\mathcal{E}_{G \leftrightarrow S_{id}}| > |\mathcal{E}_{G \leftrightarrow S_{ood}}|$, we have*
>
> $$\mathcal{M}_{sep}((A + \delta E)^2 \hat{\mathbf{g}}) > \mathcal{M}_{sep}(A^2 \hat{\mathbf{g}}),$$
>
> *where $E = \mathbf{e}_G \mathbf{e}_G^\top$ and $\delta > 0$.*

The Theorem 4.2 shows a critical principle for enhancing propagation: the optimal strategy entails the addition of edges to the subset $G$ such that there are more edges to ID data than OOD data. For some $S_{id}, S_{ood}$ in the test set, the goal is to find the set

$$G_* = \underset{S \subset \mathcal{V}_l, |S| = N_g}{\arg\max} \frac{|\mathcal{E}_{S \leftrightarrow S_{id}}|}{|\mathcal{E}_{S \leftrightarrow S_{ood}}|}, \tag{2}$$

where $N_g$ is a hyperparameter to control the size of $G_*$. Inspired by the optimization target, we proceed to present our pragmatic algorithmic approach.

## 4.2 GRAPH-AUGMENTED SCORE PROPAGATION (GRASP)

Our augmentation approach hinges on the selection of a subset, $G$, from the training set, as exemplified in Equation 2. Two principal challenges arise in implementing this: (1) We cannot directly determine the number of edges linked to ID/OOD data because these reside in the test set and their labels remain unknown. (2) An exhaustive search to find a subset is computationally expensive, as the number of combinatorial possibilities increases in a factorial manner. In this paper, we tackle these challenges by providing the practical approximation method.

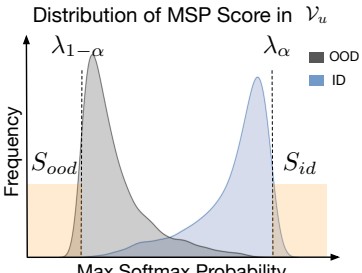

Distribution of MSP Score in $\mathcal{V}_u$

Figure 4: Illustration of the rationale in selecting $S_{id}$ and $S_{ood}$. MSP score is reported on Dataset `Coauther-CS` with the division of ID and OOD classes introduced in Appendix C.

**Selection of $\mathbf{S_{id}}/\mathbf{S_{ood}}$.** Our discussion begins by detailing the methodology to select the subset from the test ID/OOD dataset, symbolized by $S_{id}$ and $S_{ood}$ in Equation 2. A straightforward approach to obtain the most likely ID is by selecting nodes with the largest confidence and the least for OOD in class predictions. Following Hendrycks & Gimpel (2016), we employ the max softmax probability (MSP) as a representation of confidence. The selected sets can be defined as:

$$S_{id} = \{i \in \mathcal{V}_u | \max_{c \in [C]} f_c(i) > \lambda_\alpha\}, \quad S_{ood} = \{j \in \mathcal{V}_u | \max_{c \in [C]} f_c(j) < \lambda_{100-\alpha}\},$$

where $\lambda_\alpha$ denotes the $\alpha$-th percentile of the MSP scores corresponding to nodes in $\mathcal{V}_u$. To offer a clear view, Figure 4 portrays $S_{id}$ and $S_{ood}$ in the marginal regions highlighted in orange. Selecting a subset in the leftmost and rightmost regions reduces the error when identifying the ID/OOD subsets, given that overlapping between ID and OOD predominantly occurs around the central region of the distribution.

**Selection of $G$.** Upon establishing $S_{id}$ and $S_{ood}$, the next step is to determine $G$ using Equation 2. Directly enumerating every possible $G$ is impractical. Instead, we adopt a greedy approach, prioritizing the node with the highest "likelihood" score. To elucidate, for each node $i \in \mathcal{V}_l$, the score can be computed as the ratio of the edge count to $S_{id}$ over $S_{ood}$:

$$h(i) = |\mathcal{E}_{\{i\}\leftrightarrow S_{id}}|/(|\mathcal{E}_{\{i\}\leftrightarrow S_{ood}}| + 1), \tag{3}$$

where we incorporate an addition of 1 in the denominator to circumvent division by zero. Subsequently, $G$ can be expressed as:

$$G = \{i \in \mathcal{V}_l | h(i) > \tau_\beta\}, \tag{4}$$

where $\tau_\beta$ stands for the $\beta$-th percentile of $h(i)$ scores for nodes in $\mathcal{V}_l$. Once $G$ is defined, edge augmentation can be executed as demonstrated in Section 4.1. The OOD score is then propagated with the new adjacency matrix $A_+ = A + \mathbf{e}_G \mathbf{e}_G^\top$ in place:

$$\mathbf{g}_{GRASP} = (\bar{A}_+)^k \hat{\mathbf{g}}, \tag{5}$$

where $k \in \mathbb{N}^+$ are hyperparameters.

## 5 EXPERIMENTS

**Datasets.** We carry out experiments with an extensive array of graph benchmark datasets to evaluate graph OOD detection. A high-level summary of the dataset statistics is provided in Table 1, with a comprehensive description of ID/OOD split in Appendix C.1. Specifically, Cora (Sen et al., 2008) serves as a widely recognized citation network. Amazon-Photo (McAuley et al., 2015) represents a co-purchasing network on Amazon. Coauthor-CS (Sinha et al., 2015) portrays a coauthor network within the realm of computer science. Moreover,

Table 1: Summary statistics of the datasets: size of the training set $|\mathcal{V}_l|$, size of the test ID set $|\mathcal{V}_{uid}|$, size of the test OOD set $|\mathcal{V}_{uood}|$, number of ID classes $C$, scale of the dataset, and whether the graph is homophily.

| Dataset | $|\mathcal{V}_l|$ | $|\mathcal{V}_{uid}|$ | $|\mathcal{V}_{uood}|$ | $C$ | Scale | Homophily |
|---|---|---|---|---|---|---|
| Cora | 678 | 226 | 2K | 3 | SM | ✓ |
| Amazon-Photo | 2K | 1K | 4K | 3 | SM | ✓ |
| Coauthor-CS | 1K | 3K | 5K | 11 | SM | ✓ |
| Chameleon | 1K | 341 | 1K | 3 | SM | ✗ |
| Squirrel | 2K | 1K | 2K | 3 | SM | ✗ |
| ArXiv-year | 87K | 29K | 53K | 3 | LG | ✗ |
| Snap-patents | 1M | 400K | 1M | 3 | LG | ✗ |
| Wiki | 1M | 300K | 1M | 3 | LG | ✗ |

Chameleon and Squirrel (Rozemberczki et al., 2021) are two notable Wikipedia networks, predominantly utilized as heterophilic graph benchmarks. To augment the evaluation of our methods on **large-scale** graphs, we additionally incorporate three recently proposed graphs: ArXiv-year, Snap-patents, and Wiki (Lim et al., 2021).

**Remark on homophily/heterophily.** In Table 1, datasets are also categorized based on the attribute of homophily, denoting the tendency of nodes with the same class to connect. Conversely, the heterophily graph demonstrates a tendency for nodes of disparate classes to connect. This characteristic not only presents a challenge for graph classification but also for graph OOD detection. The underlying reason is that the OOD data is from different classes with ID, and heterophily exacerbates the ratio of inter-edge connections between ID and OOD, which is deemed undesirable for graph OOD detection according to Theorem 3.2.

**Implementation Details.** Our graph OOD detection technique operates in a *post hoc* fashion utilizing a pre-trained network. In particular, we explore methods with: (1) Graph Convolutional Network (GCN) (Kipf & Welling, 2017), which serves as a prototypical Graph Neural Network (GNN) model, and (2) H$_2$GCN (Zhu et al., 2020), which presents a special solution tailored to the heterophily graph learning. All pre-trained models possess a layer depth of 2. With the pre-trained network, we proceed to execute the graph OOD detection. By default, we report the performance of the augmented propagation (GRASP) on the Energy score (Liu et al., 2020). The compatibility with other OOD

Table 2: **Main results.** Comparison with competitive *post hoc* out-of-distribution detection methods. For each pre-trained method (GCN, H$_2$GCN), we take the average values that are percentages over 5 independently trained backbones. ↑ indicates larger values are better and ↓ indicates smaller values are better.

| Pre-trained Backbone | OOD Detection Method | Datasets | | | | | | | | | | Average | |
|---|---|---|---|---|---|---|---|---|---|---|---|---|---|
| | | Cora | | Amazon | | Coauthor | | Chameleon | | Squirrel | | | |
| | | FPR95 ↓ | AUROC ↑ | FPR95 ↓ | AUROC ↑ | FPR95 ↓ | AUROC ↑ | FPR95 ↓ | AUROC ↑ | FPR95 ↓ | AUROC ↑ | FPR95 ↓ | AUROC ↑ |
| GCN | MSP | 52.23 | 89.33 | 49.52 | 90.47 | 23.87 | 95.29 | 90.87 | 59.91 | 91.99 | 48.17 | 61.70 | 76.63 |
| | Energy | 52.05 | 89.48 | 39.49 | 92.33 | 14.98 | 96.52 | 94.98 | 59.68 | 94.29 | 45.06 | 59.16 | 76.61 |
| | KNN | 72.29 | 81.24 | 60.61 | 86.01 | 47.99 | 91.34 | 93.43 | 62.09 | 94.42 | **56.74** | 73.75 | 75.48 |
| | ODIN | 50.28 | 89.50 | 41.92 | 91.89 | 16.78 | 96.22 | 92.18 | 59.79 | 92.46 | 45.41 | 58.72 | 76.56 |
| | Mahalanobis | 54.89 | 88.50 | 72.63 | 83.97 | 78.39 | 87.04 | 95.15 | 48.91 | 91.46 | 55.94 | 78.50 | 72.87 |
| | GNNSAFE | 43.75 | 89.85 | **13.89** | 96.71 | 9.12 | 97.92 | 92.86 | 56.18 | 92.74 | 47.08 | 50.47 | 77.55 |
| | **GRASP (ours)** | **21.92** | **94.65** | 15.64 | **96.76** | **7.88** | **97.94** | 74.54 | **67.97** | **90.21** | 54.93 | **42.04** | **82.45** |
| H$_2$GCN | MSP | 54.19 | 90.57 | 71.55 | 84.69 | 50.22 | 90.66 | 85.87 | 68.43 | 92.74 | 52.36 | 70.91 | 77.34 |
| | Energy | 42.94 | 91.71 | 56.60 | 85.92 | 43.97 | 92.27 | 91.57 | 66.03 | 92.92 | 47.13 | 65.60 | 76.61 |
| | KNN | 65.89 | 86.45 | 56.55 | 84.94 | 50.45 | 91.43 | 92.34 | 61.73 | 94.35 | **61.14** | 71.92 | 77.14 |
| | ODIN | 41.39 | 91.45 | 60.04 | 86.57 | 47.58 | 91.80 | 90.02 | 69.06 | 91.23 | 53.97 | 66.05 | 78.57 |
| | Mahalanobis | 77.34 | 84.34 | 95.82 | 73.12 | 64.33 | 87.90 | 97.71 | 57.37 | 97.71 | 57.37 | 86.58 | 72.02 |
| | GNNSAFE | 35.11 | 93.56 | 20.22 | 95.64 | 33.47 | 93.48 | 89.59 | 62.61 | 92.21 | 45.29 | 54.12 | 78.12 |
| | **GRASP (ours)** | **18.30** | **95.65** | **14.50** | **96.53** | **8.29** | **97.59** | 65.35 | **72.63** | **90.06** | 56.00 | **39.30** | **83.68** |

scoring functions is also shown in Table 3. We set the propagation number $k$ as 8, with percentile values $\alpha = 5$ and $\beta = 50$.

**Metrics.** Following the convention in literature (Hendrycks & Gimpel, 2016; Liu et al., 2020; Sun et al., 2021), we use AUROC and FPR95 as evaluation metrics for OOD detection.

## 5.1 COMPARATIVE RESULTS

**GRASP achieves superior performance.** We provide results in Table 2, wherein our proposed methodology (GRASP) demonstrates promising performance. The comparative analysis encompasses a broad spectrum of *post hoc* competitive Out-of-Distribution (OOD) detection techniques in existing literature. We categorize the baseline methods into two groups: (a) Traditional OOD detection methods including MSP (Hendrycks & Gimpel, 2016), Energy (Liu et al., 2020), ODIN (Liang et al., 2018), and KNN (Sun et al., 2022a); (b) Graph OOD detection methods GNNSAFE (Wu et al., 2022). In this table, we present GRASP results based on the Energy score. Noteworthy findings include: (a) The traditional OOD detection methods exhibit suboptimal performance in the realm of graph OOD detection. For instance, GRASP reduced the average FPR95 by **26.30**% compared to the strongest traditional OOD detection method (Energy) with H$_2$GCN. This outcome is anticipated given their lack of specificity in design towards graph data. (b) GRASP outperforms existing baselines by a large margin, surpassing the best baseline GNNSAFE by **14.82**% concerning average FPR95 on the pre-trained backbone H$_2$GCN. These results further corroborate that the theoretically motivated solution GRASP is also appealing to use in practice.

Table 3: GRASP is compatible with different OOD scoring functions. We compare OOD detection methods and the performance after the simple propagation in Equation 1 (denoted by "+ prop") and with GRASP respectively. We report FPR95 results that are averaged over 5 independent models pre-trained with GCN and 5 models pre-trained with H$_2$GCN.

| Method | DATASET | | | | | |
|---|---|---|---|---|---|---|
| | Cora | Amazon | Coauthor | Chameleon | Squirrel | Average |
| MSP | 53.21 | 60.54 | 37.05 | 88.37 | 92.37 | 66.31 |
| MSP + prop | 30.04 | 29.06 | 20.38 | 91.42 | 91.46 | 52.47 |
| MSP + GRASP (Ours) | **20.21** | **19.89** | **7.75** | **75.11** | **90.61** | **42.71** |
| Energy | 47.50 | 48.05 | 29.48 | 93.28 | 93.61 | 62.38 |
| Energy + prop | 30.33 | 22.54 | 27.55 | 97.74 | 90.74 | 53.78 |
| Energy + GRASP (Ours) | **20.11** | **15.07** | **8.09** | **69.95** | **90.14** | **40.67** |
| KNN | 69.09 | 58.58 | 49.22 | 92.89 | 94.39 | 72.83 |
| KNN + prop | 46.67 | 33.84 | 15.23 | 90.12 | 92.00 | 55.57 |
| KNN + GRASP (Ours) | **32.49** | **20.05** | **8.59** | **66.52** | **88.43** | **43.21** |

**GRASP is compatible with a wide range of OOD scoring methods.** In Table 3, we demonstrate the compatibility of GRASP with various alternative scoring functions. We evaluate commonly utilized scoring functions, comparing the performance with and without the application of GRASP accordingly. We specifically examine MSP (Hendrycks & Gimpel, 2016), Energy (Liu et al., 2020),

and KNN (Sun et al., 2022a), each of which generates OOD scores to form a scoring vector; GRASP is then applied to facilitate score propagation. Notably, across all five datasets, the use of GRASP markedly surpasses the performance of its non-augmented counterpart. The results, as presented in Table 3, are averaged across all backbones (GCN and H$_2$GCN). *The detailed performance on each backbone is shown in Appendix, Table 9 and Table 10 .*

**GRASP is better than propagation without augmentation.** In Table 3, we contrast the performance of two propagation strategies including the basic propagation illustrated in Equation 1, and the enhanced propagation strategy, GRASP, as introduced in Equation 5. Our observations indicate that utilizing basic propagation with the original graph connection may result in diminished performance. For instance, within the Chameleon dataset, there is an increase in the FPR95 by 3.04% and 4.46% respectively on MSP and Energy scores. This outcome resonates with the discussion in Section 3, affirming that propagation does not always enhance graph OOD detection. To boost the post-propagation OOD detection performance, we suggest employing augmented propagation and the empirical results demonstrate that GRASP consistently outperforms the basic propagation strategy.

**GRASP is also competitive on large-scale graph datasets.** We further extend our evaluation to the large-scale graph OOD detection task, leveraging datasets such as ArXiv-year, Snap-patents, and Wiki (Lim et al., 2021). Contrasted with the small-scale benchmarks in Table 4, the large-scale scenario presents more challenges due to a large number of nodes and edges. Through empirical analysis, we find that all baseline OOD detection methodologies exhibit suboptimal performance, manifesting around 50%

Table 4: Results on large-scale graph dataset. We report AUROC that are averaged over 5 independently pre-trained GCN models.

| Method | Large-scale Dataset | | | |
|---|---|---|---|---|
| | ArXiv-year | Snap-patents | Wiki | Average |
| MSP | 43.35 | 51.42 | 36.63 | 43.80 |
| Energy | 47.58 | 46.93 | 28.47 | 40.99 |
| KNN | 61.28 | 53.20 | 40.89 | 51.79 |
| ODIN | 43.62 | 49.09 | 34.03 | 42.25 |
| Mahalanobis | 59.61 | 55.29 | 61.06 | 58.65 |
| GNNSAFE | 36.66 | 33.44 | 39.90 | 36.67 |
| **GRASP (ours)** | **74.66** | **67.36** | **65.56** | **69.19** |

AUROC. This outcome aligns with anticipations, given that these three datasets are recognized for their heterophily graph characteristic. By deploying our augmentation strategy, we manage to elevate the overall AUROC to approximately 70%, marking a substantial improvement of **10.54%** over the second-best methodology.

**Comparison with training-based graph OOD detection baselines.** In addition to contrasting with *post hoc* methods, we extend our comparison to a parallel line of graph Out-Of-Distribution (OOD) detection research, which focuses on refining the training strategy to improve graph OOD detection performance. The methods compared include OE (Hendrycks et al., 2018), GKDE (Zhao et al., 2020), and GPN (Stadler et al., 2021). While these approaches necessitate a costly re-training procedure, our approach, GRASP, offers a simple "plug-and-play" utility on any pre-trained models, and furthermore demonstrates superior performance compared to the baseline methods, as illustrated in Table 5.

Table 5: Comparison with training-based OOD detection baselines on AUROC.

| Method | Dataset | | |
|---|---|---|---|
| | Cora | Amazon | Coauther |
| OE | 89.47 | 95.39 | 96.04 |
| GKDE | 57.23 | 65.58 | 61.15 |
| GPN | 90.34 | 92.72 | 83.65 |
| **GRASP (ours)** | **94.65** | **96.76** | **97.94** |

## 5.2 FURTHER DISCUSSIONS

**Selection by $h(i)$ is effective.** The essence of our approach, GRASP, lies in selecting a subset $G$ from the training set such that there is a higher edge count towards ID data compared to OOD data (Equation 2). Our objective is to find out whether the estimation score $h(i)$, detailed in Equation 3, can effectively prioritize a training node that has a higher edge count to ID ($\mathcal{V}_{uid}$) over OOD ($\mathcal{V}_{uid}$). Figure 5 demonstrates the sorting of training nodes indices from low to high, revealing that a higher $h(i)$ value (corresponding to the right side of the figure) is associated with more edges towards ID than OOD, thereby affirming the efficacy of our algorithm.

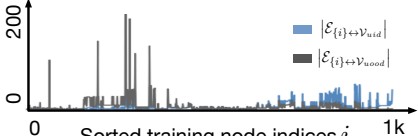

Figure 5: Illustration of the number of edges from each training node $i$ to ID and OOD data within the test set of the Chameleon dataset. The x-axis denotes the training node indices, ordered by $h(i)$ from low to high.

**Selecting the subset G strategically is important.** In our GRASP algorithm, we select the subset $G$ from the training set, comprising nodes with the top 50% scores of $h(i)$, which correspond to the nodes on the right side of Figure 5. Upon altering the selection policy to include 50% of nodes with the lowest $h(i)$ values (left side of Figure 5), the AUROC declines from 67.97% to 60.46% on the GCN backbone. This result substantiates the theoretical insight posited in Theorem 4.2, affirming that selecting subset $G$ with more edges to ID data than OOD data can enhance graph OOD detection performance after the augmented propagation.

**Directly adding edges to $S_{id}$ and $S_{ood}$ is sub-optimal.** In this paragraph, we draw a comparison with an alternative solution to GRASP. While we augment the training set subset with additional edges, there is also a possibility of directly incorporating edges into $S_{id}$ and $S_{ood}$ within the test set. Employing this strategy yields an average AUROC on GCN of 72.44, which, according to Table 2, is approximately 10% lower than that achieved with GRASP. This observation further substantiates the notion that "confirmation bias" can adversely affect the graph OOD detection.

## 6 RELATED WORK

**Out-of-distribution Detection.** The primary focus within this realm has been on the development of scoring functions for OOD detection. These works can be broadly categorized into two main streams: (1) output-based methods (Hendrycks & Gimpel, 2016; Liang et al., 2018; Liu et al., 2020; Wang et al., 2022a; Huang et al., 2021; Wang et al., 2022b; Zhu et al., 2022; Huang et al., 2022; Djurisic et al., 2022; Zhang et al., 2023), and (2) feature-based methods including the Mahalanobis distance (Lee et al., 2018; Sehwag et al., 2021; Ren et al., 2021) and KNN distance (Sun et al., 2022a). These methodologies are predominantly applied in domains such as computer vision, where samples are inherently independent of each other. However, these techniques are not designed to adeptly handle data structures like graphs, where samples are inter-connected.

**Out-of-distribution detection for graph data.** Graph anomaly detection has a rich history (Ding et al., 2021a; Wang et al., 2021b; Zhang et al., 2021; Liu et al., 2021a; Wang et al., 2021c; Liu et al., 2021c; Ding et al., 2021b; Liu et al., 2021d; Kim et al., 2022). In recent years, the OOD detection in graph data introduced fresh challenges, particularly with multi-class classification for in-distribution data, escalating the difficulty in discerning outlier data. Some of the works focus on graph-level OOD detection (Li et al., 2022; Liu et al., 2023; Bazhenov et al., 2022). For node-level OOD detection, GKDE (Zhao et al., 2020) and GPN (Stadler et al., 2021) apply Bayesian Network models to estimate uncertainties to detect OOD nodes. However, Bayesian-based approaches can encounter impediments such as inaccurate predictions and high computational demands, which limit their broader applicability Yang et al. (2021). GNNSAFE (Wu et al., 2022) emerges as the work employing post hoc energy-based score to perform OOD detection. Given the merits of post hoc methods, our study first provides a comprehensive understanding of the OOD score propagation in Graphs, extending beyond existing knowledge.

**Graph Data Augmentation.** Graph Data Augmentation is a common technique in graph machine learning (Gasteiger et al., 2019b; Chen et al., 2020; Rong et al., 2020; Jin et al., 2020; Zheng et al., 2020; Zhao et al., 2021; Kipf & Welling, 2016; Park et al., 2021; Ding et al., 2022; Azabou et al., 2023) to improve the node classification performance. Existing methods operate exclusively on in-distribution (ID) data. Furthermore, their test set data also originates from the in-distribution and shares the same classes as the training set. In contrast, our data augmentation is purposefully crafted for OOD detection, supported by the theoretical explanation.

## 7 CONCLUSION

In this research, we delve into an important yet under-explored challenge in the realm of graph data: Out-of-Distribution (OOD) detection. Recognizing the inadequacies of traditional OOD detection techniques in the context of graph data, our exploration centered on the potential of score propagation as a viable and efficient solution. Our findings reveal the specific conditions under which score propagation will be helpful—in situations where the ratio of intra-edges surpasses that of inter-edges. Motivated by this finding, our edge augmentation strategy selectively adds edges to a specific subset $G$ of the training set, which provably improves post-propagation OOD detection outcomes under certain conditions. Extensive empirical evaluations reinforced the merit of our approach. In summary, this paper contributes an enriched understanding of OOD detection in graph data and paves the way for more robust graph-based machine learning systems.

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

# A    DETAILS OF BASELINES

For the reader's convenience, we summarize in detail a few common techniques for defining OOD scores that measure the degree of ID-ness on a given input. By convention, a higher (lower) score is indicative of being in-distribution (out-of-distribution).

**MSP Hendrycks & Gimpel (2016)** This method proposes to use the maximum softmax score as the OOD score. For each node $i$, we use $F_{OODD}(i) = \max_{c \in [C]} f_c(i)$ as the OOD score.

**ODIN Liang et al. (2018)** This method improves OOD detection with temperature scaling and input perturbation. In all experiments, we set the temperature scaling parameter $T = 1000$. For graph neural network, we found the input perturbation does not further improve the OOD detection performance and hence we set $\epsilon = 0$.

**Mahalanobis Lee et al. (2018)** This method uses multivariate Gaussian distributions to model class-conditional distributions of softmax neural classifiers and uses Mahalanobis distance-based scores for OOD detection. The mean $\mu_c$ of each multivariate Gaussian distribution with class $c$ and a tied covariance $\Sigma$ are estimated based on training samples. We define the confidence score $M(\mathbf{x})$ using the Mahalanobis distance between test sample $\mathbf{x}$ and the closest class-conditional Gaussian distribution.

**Energy Liu et al. (2020)** This method proposes using energy score for OOD detection. The energy function maps the logit outputs to a scalar $E(\mathbf{x}_i; f) \in \mathbb{R}$, which is relatively lower for ID data. Note that Liu et al. (2020) used the *negative energy score* for OOD detection, in order to align with the convention that $S(\mathbf{x})$ is higher (lower) for ID (OOD) data.

**KNN Sun et al. (2022a)** This method uses the $k$-th nearest neighbor distance between a test graph node and the training set as the OOD score. We use $k = 10$ for all experiments in this paper.

# B    TECHNICAL DETAILS

**Theorem B.1.** *(Recap of Theorem 3.2) For any two test ID/OOD node set $S_{id} \subset \mathcal{V}_{uid}, S_{ood} \subset \mathcal{V}_{uood}$ with equal size $N_s$, let the ID-vs-OOD separability $\mathcal{M}_{sep}$ defined on a OOD scoring vector $\hat{\mathbf{g}} \in \mathbb{R}^N$ as*

$$\mathcal{M}_{sep}(\hat{\mathbf{g}}) \triangleq \mathbb{E}_{i \in \mathcal{S}_{id}} \hat{\mathbf{g}}_i - \mathbb{E}_{j \in \mathcal{S}_{ood}} \hat{\mathbf{g}}_j.$$

*If $\mathcal{M}_{sep}(\hat{\mathbf{g}}) > 0$ and $\eta_{intra} - \eta_{inter} > 1/N_s$, for some $\epsilon > 0$ and constant $c$, we have*

$$\mathbb{P}\left(\mathcal{M}_{sep}(A\hat{\mathbf{g}}) \geq \mathcal{M}_{sep}(\hat{\mathbf{g}}) - \epsilon\right) \geq 1 - exp(-\frac{c\epsilon^2}{\|\hat{\mathbf{g}}\|_2^2}).$$

*Proof.* Without losing the generality, we set the $\hat{\mathbf{g}}_i = 0$, if $i \in S_{ood} \cup S_{id}$, since we only care about the detection results in the given test node set $S_{ood}$ and $S_{id}$.

The $\mathcal{M}_{sep}(\hat{\mathbf{g}})$ can be re-written as

$$\mathcal{M}_{sep}(\hat{\mathbf{g}}) = \hat{\mathbf{g}}^\top (\mathbf{e}_{S_{id}} - \mathbf{e}_{S_{ood}}).$$

Then we have

$$\mathcal{M}_{sep}(A\hat{\mathbf{g}}) = \hat{\mathbf{g}}^\top A(\mathbf{e}_{S_{id}} - \mathbf{e}_{S_{ood}})$$

According to General Hoeffding's inequality (Theorem 2.6.3) in Vershynin (2018), we know that

$$\mathbb{P}\left(\mathbb{E}[\hat{\mathbf{g}}^\top A(\mathbf{e}_{S_{id}} - \mathbf{e}_{S_{ood}})] - \hat{\mathbf{g}}^\top A(\mathbf{e}_{S_{id}} - \mathbf{e}_{S_{ood}}) \leq \epsilon\right) \geq 1 - \exp(-\frac{c\epsilon^2}{\|\hat{\mathbf{g}}\|_2^2}),$$

where $c$ is some constant value.

Since $\hat{\mathbf{g}}_i = 0$, if $i \in \mathcal{V}_l$,

$$
\begin{aligned}
\mathbb{E}[\hat{\mathbf{g}}^\top A(\mathbf{e}_{S_{id}} - \mathbf{e}_{S_{ood}})] &= \hat{\mathbf{g}}^\top \mathbb{E}[A](\mathbf{e}_{S_{id}} - \mathbf{e}_{S_{ood}}) \\
&= \hat{\mathbf{g}}^\top N_s(\eta_{intra} - \eta_{inter})(\mathbf{e}_{S_{id}} - \mathbf{e}_{S_{ood}}) \\
&> \hat{\mathbf{g}}^\top (\mathbf{e}_{S_{id}} - \mathbf{e}_{S_{ood}})
\end{aligned}
$$

Combining together, we have

$$
\mathbb{P}\left(\hat{\mathbf{g}}^\top A(\mathbf{e}_{S_{id}} - \mathbf{e}_{S_{ood}}) \geq \hat{\mathbf{g}}^\top(\mathbf{e}_{S_{id}} - \mathbf{e}_{S_{ood}}) - \epsilon\right) \geq 1 - \exp(-\frac{c\epsilon^2}{\|\hat{\mathbf{g}}\|_2^2})
$$

$\square$

**Theorem B.2.** *(Recap of Theorem 4.2) For any two test ID/OOD node set $S_{id} \subset \mathcal{V}_{uid}$, $S_{ood} \subset \mathcal{V}_{uood}$ with equal size $N_s$, let the ID-vs-OOD separability $\mathcal{M}_{sep}$ defined on a non-negative OOD scoring vector $\hat{\mathbf{g}} \in \mathbb{R}^N$ as*

$$
\mathcal{M}_{sep}(\hat{\mathbf{g}}) \triangleq \mathbb{E}_{i \in \mathcal{S}_{id}} \hat{\mathbf{g}}_i - \mathbb{E}_{j \in \mathcal{S}_{ood}} \hat{\mathbf{g}}_j.
$$

*Let $\mathcal{E}_{S \leftrightarrow S'} \subset \mathcal{E}$ to denote the edge set of edges between two node sets $S$ and $S'$, where $S, S' \subset \mathcal{V}$. If we can find a node set $G \subset \mathcal{V}_l$ such that $|\mathcal{E}_{G \leftrightarrow S_{id}}| > |\mathcal{E}_{G \leftrightarrow S_{ood}}|$, we have*

$$
\mathcal{M}_{sep}((A + \delta E)^2 \hat{\mathbf{g}}) > \mathcal{M}_{sep}(A^2 \hat{\mathbf{g}}),
$$

*where $E = \mathbf{e}_G \mathbf{e}_G^\top$ and $\delta > 0$.*

*Proof.* The $\mathcal{M}_{sep}(\hat{\mathbf{g}})$ can be re-written as

$$
\mathcal{M}_{sep}(\hat{\mathbf{g}}) = \frac{1}{N_s} \hat{\mathbf{g}}^\top (\mathbf{e}_{S_{id}} - \mathbf{e}_{S_{ood}}).
$$

We can then directly derive the proof by expanding

$$
\begin{aligned}
\mathcal{M}_{sep}((A + \delta E)^2 \hat{\mathbf{g}}) - \mathcal{M}_{sep}(A^2 \hat{\mathbf{g}}) &= \frac{1}{N_s}(\mathbf{e}_{S_{id}} - \mathbf{e}_{S_{ood}})^\top \left((A + \delta E)^2 \hat{\mathbf{g}} - A^2 \hat{\mathbf{g}}\right) \\
&= \frac{1}{N_s}(\mathbf{e}_{S_{id}} - \mathbf{e}_{S_{ood}})^\top \left(\delta(AE + EA)\hat{\mathbf{g}} + \delta^2 \mathbf{e}_G \mathbf{e}_G^\top \mathbf{e}_G \mathbf{e}_G^\top \hat{\mathbf{g}}\right) \\
&= \frac{\delta}{N_s}(\mathbf{e}_{S_{id}} - \mathbf{e}_{S_{ood}})^\top \left(A\mathbf{e}_G \mathbf{e}_G^\top \hat{\mathbf{g}} + \mathbf{e}_G \mathbf{e}_G^\top A\hat{\mathbf{g}}\right) \\
&= \frac{\delta}{N_s}(\mathbf{e}_{S_{id}} - \mathbf{e}_{S_{ood}})^\top A\mathbf{e}_G \mathbf{e}_G^\top \hat{\mathbf{g}} \\
&= \frac{\delta}{N_s}(\mathbf{e}_G^\top \hat{\mathbf{g}})(\mathbf{e}_{S_{id}}^\top A\mathbf{e}_G - \mathbf{e}_{S_{ood}}^\top A\mathbf{e}_G) \\
&= \frac{\delta(\mathbf{e}_G^\top \hat{\mathbf{g}})}{|G| N_s}(|\mathcal{E}_{G \leftrightarrow S_{id}}| - |\mathcal{E}_{G \leftrightarrow S_{ood}}|) \\
&> 0,
\end{aligned}
$$

where the second and the third equation are derived by the fact that $G \subset \mathcal{V}_l$ and then we have $\mathbf{e}_{S_{id}}^\top \mathbf{e}_G = 0$ and $\mathbf{e}_{S_{ood}}^\top \mathbf{e}_G = 0$.

$\square$

# C EXPERIMENTAL DETAILS

## C.1 DATASET DETAILS

We adopt nine publicly available common benchmarks used for graph learning, split the ID/OOD parts and use them in our experiment. For `Cora`, `Amazon-Photo` and `Chameleon`, we follow the approaches handling these datasets as (Wu et al., 2022) and use the data loader provided by

the Pytorch Geometric package [1]. For the remaining six datasets, we directly use the pickle file or download from the given hyperlinks proposed by Lim et al. (2021).

Cora (Sen et al., 2008) is a 7-class citation network comprising 2,708 nodes, 5,429 edges and 1,433 features. In this network, each node represents a published paper, each edge signifies a citation relationship, and the label class is each paper's topic, which is the goal to predict. For our OOD setting, we designate nodes belonging to 4 specific classes (0, 1, 2, 3) as Out-of-Distribution (OOD), while the remaining 3 classes (4, 5, 6) are considered In-Distribution (ID).

Amazon-Photo (McAuley et al., 2015) is an 8-class item co-purchasing network on Amazon, which contains 7,650 nodes, 238,162 edges and 745 features. In this network, each node denotes a product, each edge indicates that two linked products are frequently purchased together, and the node label denotes the category of the product. Similar to Cora, we designate 3 classes (5, 6, 7) as ID and and the remaining 5 classes (0, 1, 2, 3, 4) as Out-of-Distribution (OOD).

Coauthor-CS (Sinha et al., 2015) is a 15-class coauthor network of computer science, which contains 18,333 nodes, 163,788 edges and 6,805 features. In this network, nodes denote authors and there is an edge between two authors if co-authored a paper. And the label represents the study field for the authors. Similar to Cora, we use 4 classes (0, 1, 2, 3) of nodes as Out-of-Distribution (OOD), and the remaining 10 classes (4-14) are used as In-Distribution (ID).

The data split operation for ID and OOD in the aforementioned three datasets is the same as GNNSAFE (Wu et al., 2022). Notably, GNNSAFE offers three OOD processing methods for these datasets, of which, however, only the "Label Leave-Out" approach is the correct setup for OOD detection. Furthermore, the methodologies applied to other datasets in GNNSAFE do not align with node-level graph OOD detection setup either, by which reason we only adopt these three datasets.

Chameleon and Squirrel (Rozemberczki et al., 2021) are two Wikipedia networks with 5 classes, where nodes represent web pages and edges represent hyperlinks between them. Node features represent several informative nouns in the Wikipedia pages and the task is to predict the average daily traffic of the web page (Fey & Lenssen, 2019).

arXiv-year (Hu et al., 2020) is the ogbn-arXiv network with different labels and is altered to be heterophily, in which the class labels are set to be the year that the paper is posted, instead of subject area in the original paper. The nodes are arXiv papers, and directed edges connect a paper to other papers that it cites. The node features are averaged word2vec token features of both the title and abstract of the paper. The five classes are chosen by partitioning the posting dates so that class ratios are approximately balanced (Lim et al., 2021).

snap-patents (Leskovec & Krevl, 2014; Leskovec et al., 2005) is a big dataset of utility patents in the US. Each node represents a patent and edges connect patents that cite each other. Node features are derived from patent metadata (Lim et al., 2021). Like arXiv-year, this dataset is changed to set the task to predict the time at which a patent was granted, which is also five classes.

wiki (Lim et al., 2021) is a super big dataset of Wikipedia articles, which are crawled and cleaned from the internet. Nodes represent pages and edges represent links between them. Node features are derived from the average GloVe embeddings (Pennington et al., 2014) of the titles and abstracts and labels indicate total page views over a 60-day period, categorized into five classes based on quintiles.

The above 6 datasets all have 5 classes, for which we universally designate three classes (2, 3, 4) as ID and the remaining 2 classes (0, 1) as OOD, just as Cora, Amazon-Photo and Coauthor-CS.

The complete information and statistics of all these datasets aforementioned are summarized in Table 6.

## C.2 GRAPH CLASSIFICATION DETAILS

The ID Accuracy result of the pre-trained backbone model is shown in Tables 7 and 8.

---

[1]https://pytorch-geometric.readthedocs.io/en/latest/modules/datasets.html

Table 6: Statistics of all the graph datasets. # C is the total number of distinct node classes.

| Dataset | # Nodes | # Edges | # Features | # C | OOD Class | ID Class |
|---|---|---|---|---|---|---|
| Cora | 2,708 | 5,429 | 1,433 | 7 | {0, 1, 2, 3} | {4,5,6} |
| Amazon-Photo | 7,650 | 238,162 | 745 | 8 | {0, 1, 2, 3, 4} | {5, 6, 7} |
| Coauthor-CS | 18,333 | 163,788 | 6,805 | 15 | {0, 1, 2, 3} | {4, $\cdots$, 14} |
| Chameleon | 2,277 | 31,421 | 2,325 | 5 | {0, 1} | {2, 3, 4} |
| Squirrel | 5,201 | 198,493 | 2,089 | 5 | {0, 1} | {2, 3, 4} |
| arXiv-year | 169,343 | 1,166,243 | 128 | 5 | {0, 1} | {2, 3, 4} |
| snap-patents | 2,923,922 | 13,975,788 | 269 | 5 | {0, 1} | {2, 3, 4} |
| wiki | 1,925,342 | 303,434,860 | 600 | 5 | {0, 1} | {2, 3, 4} |

Table 7: ID ACCs of five small-scale graph datasets.

| Dataset | Cora | Amazon | Coauthor | Chameleon | Squirrel |
|---|---|---|---|---|---|
| GCN | $93.894 \pm 1.305$ | $96.723 \pm 0.761$ | $96.214 \pm 0.610$ | $71.202 \pm 2.067$ | $72.318 \pm 2.015$ |
| $H_2$GCN | $94.071 \pm 1.840$ | $96.258 \pm 0.741$ | $94.565 \pm 0.433$ | $71.848 \pm 2.214$ | $74.725 \pm 1.107$ |

Table 8: ID ACCs of three large-scale graph datasets.

| Dataset | arXiv-year | snap-patents | wiki |
|---|---|---|---|
| GCN | $56.67 \pm 0.33$ | $62.48 \pm 0.10$ | $54.89 \pm 0.16$ |

Table 9: GRASP is compatible with different OOD scoring functions. We compare OOD detection methods and the performance after the simple propagation in Equation 1 (denoted by "+ prop") and with GRASP respectively. We report FPR95 results that are averaged over 5 independent models pre-trained with GCN.

| Method | DATASET | | | | | |
|---|---|---|---|---|---|---|
| | Cora | Amazon | Coauthor | Chameleon | Squirrel | Average |
| MSP | 52.23 | 49.52 | 23.87 | 90.87 | 91.99 | 61.70 |
| MSP + prop | 31.24 | 26.87 | 8.74 | 98.17 | 91.77 | 51.36 |
| MSP + GRASP (Ours) | **19.31** | **18.38** | **7.31** | **94.10** | **91.16** | **46.05** |
| Energy | 52.05 | 39.49 | 14.98 | 94.98 | 94.29 | 59.16 |
| Energy + prop | 38.03 | 23.42 | 11.86 | 97.77 | 91.00 | 52.42 |
| Energy + GRASP (Ours) | **21.92** | **15.64** | **7.88** | **74.54** | **90.21** | **42.04** |
| KNN | 72.29 | 60.61 | 47.99 | 93.43 | 94.42 | 73.75 |
| KNN + prop | 49.94 | 38.95 | 15.05 | 92.71 | 92.48 | 57.83 |
| KNN + GRASP (Ours) | **33.16** | **21.43** | **8.86** | **62.29** | **87.96** | **42.74** |

Table 10: GRASP is compatible with different OOD scoring functions. We compare OOD detection methods and the performance after the simple propagation in Equation 1 (denoted by "+ prop") and with GRASP respectively. We report FPR95 results that are averaged over 5 independent models pre-trained with $H_2$GCN.

| Method | DATASET | | | | | |
|---|---|---|---|---|---|---|
| | Cora | Amazon | Coauthor | Chameleon | Squirrel | Average |
| MSP | 54.19 | 71.55 | 50.22 | 85.87 | 92.74 | 70.91 |
| MSP + prop | 28.84 | 31.24 | 32.01 | 84.67 | 91.14 | 53.58 |
| MSP + GRASP (Ours) | **21.11** | **21.39** | **8.18** | **56.11** | **90.05** | **39.37** |
| Energy | 42.94 | 56.60 | 43.97 | 91.57 | 92.92 | 65.60 |
| Energy + prop | 22.63 | 21.65 | 43.23 | 97.71 | 90.47 | 55.14 |
| Energy + GRASP (Ours) | **18.30** | **14.50** | **8.29** | **65.35** | **90.06** | **39.30** |
| KNN | 65.89 | 56.55 | 50.45 | 92.34 | 94.35 | 71.92 |
| KNN + prop | 43.40 | 28.73 | 15.41 | 87.53 | 91.52 | 53.32 |
| KNN + GRASP (Ours) | **31.81** | **18.67** | **8.32** | **70.74** | **88.89** | **43.69** |

