# OpenReview forum: "Score Propagation as a Catalyst for Graph Out-of-distribution Detection: A Theoretical and Empirical Study"
_ICLR.cc/2024/Conference — Submitted to ICLR 2024_

### Official Review · Reviewer_13zH · 2023-10-27

**Soundness:** 3 good
**Presentation:** 3 good
**Contribution:** 2 fair
**Rating:** 6
**Confidence:** 3

**Summary:**

The paper proposes a method called GRaph-Augmented Score Propagation (GRASP) for enhancing out-of-distribution (OOD) detection in graphs. It propagates OOD scores among neighboring nodes to leverage graph structure. The paper investigates whether score propagation will always help graph OOD detection. The authors find the ratio of intra-edges (ID-ID and OOD-OOD) to inter-edges (ID-OOD) must be high for propagation to be beneficial. To improve this ratio, GRASP strategically adds edges to a subset G of training nodes that are assured to be in-distribution. This enhances the intra-edge ratio and thus OOD detection performance after propagation. Theoretically, the paper shows that if G connects predominantly to ID data over OOD data, GRASP can provably improve post-propagation OOD detection outcomes. The paper evaluates GRASP on benchmark graph datasets and pre-trained GNNs. It demonstrates GRASP outperforms baselines
Overall, the paper is well-written and provides good insight. However, the experiment was all conducted on small-scale of data, which  limits the ability to fully validate the proposed approach and conclusions. More experiments on large-scale real world data would strength the claims.

**Strengths:**

1. The paper provides one of the first theoretical analyses of score propagation for graph OOD detection. The theoretical analysis is derived rigorously and proves helpful conditions for when propagation enhances OOD detection. The formulations and proofs are clear.
2. The paper is well-written, the motivation, problem definition, methodology and conclusions are explained clearly throughout the paper.
3. The proposed GRASP method is original in its strategic augmentation of edges to a subset of training nodes to boost intra-edge ratios.

**Weaknesses:**

1. However, the experiment was all conducted on small-scale of data, which  limits the ability to fully validate the proposed approach and conclusions. More experiments on large-scale real world data would strength the claims.
2. The time complexity of GRASP compared to baselines is not mentioned in the paper. A thorough accounting of computation/memory demands compared to baselines is important as it relates to practical deployment.

**Questions:**

1. The theoretical analysis assumes edges follow a Bernoulli distribution. How sensitive are the results to this assumption?
2. How do different propagation mechanisms, like higher-order diffusion, impact the findings?
3. What is the time complexity of GRASP compared to baselines?

---

> ### Author Response · Authors · 2023-11-18
> **Response - Part 1**
>
> We thank the reviewer for the insightful questions! Below we address each of your comments in detail.
>
> > **Q1. Experiments on large-scale real world data.**
>
> We are glad to share with you that we have conducted experiments on large datasets in paper, i.e. arxiv-year, snap-patents, and wiki, which are three large-scale real world datasets reflecting real-world scenarios. Specifically, Arxiv-year has node and edge counts in the order of hundreds of thousands and millions, respectively. Snap-patents features millions of nodes and tens of millions of edges, while wiki, being a large dense graph, boasts millions of nodes and billions of edges. The detailed statistics of these datasets [1] are provided in the following table. We believe these datasets sufficiently meet the requirements of real-world applications. If the reviewer has an interest in other large datasets, feel free to share with us and we would be more than willing to conduct additional experiments on these datasets to further check the effectiveness of our proposed method.
>
> |Dataset | #Nodes | #Edges |
> |:-------- |--------:|--------:|
> | arXiv-year | 169,343 | 1,166,243 |
> |snap-patents |**2,923,922** |13,975,788 |
> |wiki|**1,925,342**|**303,434,860**|
>
> [1] Derek Lim, Felix Hohne, Xiuyu Li, Sijia Linda Huang, Vaishnavi Gupta, Omkar Bhalerao and Ser-Nam Lim. Large Scale Learning on Non-Homophilous Graphs: New Benchmarks and Strong Simple Methods. NeurIPS 2021.
>
>
>
> > **Q2. The time complexity of GRASP and a thorough accounting of computation/memory demands compared to baselines to account for practical deployment.**
>
> Great question! We first present the conclusion and then provide the support from both mathmatical and empirical points.
>
> **Takeway:** Our method's computational footprint in terms of a single propagation’s runtime and memory usage is:
> - Time Complexity: $O(N+|\mathcal{E}|+n)$,
> - Memory Complexity: $O(N+|\mathcal{E}|+n)$,
>
> where $N$ is node count of the entire graph $\mathcal{G}$ with $|\mathcal{E}|$ edges and subset $G$ has $n$ nodes. Hence, our algorithm has a linear complexity and this makes our method applicable to large-scale networks.
>
> **Justification:** While our algorithm introduces the fully connected matrix $E$, we adeptly transform the propagation formula to eliminate direct matrix computation between $E$ and the OOD score vector $\mathbf{g}$, resulting in an efficient linear complexity. This can be  confirmed by the computational complexity  analysis presented below.
> - **Computational complexity analysis.** As described by Equation (5) in our paper, given a raw OOD scoring vector $\hat{\mathbf{g}} \in \mathbb{R}^{N}$, the propagated scoring vector using augmentated adjacency matrix after propagation for $k$ times is given by:
>  \begin{align*}
>      \mathbf{g} \_{GRASP} &= (\bar{A} \_+)^k \hat{\mathbf{g} }
>      = ({D}^{-1} \_+A \_+)^k \hat{\mathbf{g}} = ({D}^{-1} \_+(A+E))^k \hat{\mathbf{g}} \\
>      &= ({D}^{-1} \_+A \_+)^{k-1} \cdot (D \_+^{-1}\boxed{A\hat{\mathbf{g}}}+D \_+^{-1}\boxed{E\hat{\mathbf{g}}})   \\
>      &= (\text{run $k-1$ times ...}) \\
> \end{align*}
> In the above equation, ($a$) $E\hat{\mathbf{g}}$ can be computed with time/space complexity $O(n)$ by a simple summation operation ($\mathbf{g} \_i$ in $G$ will be replaced by $\sum \_{i\in G} \mathbf{g} \_i$ ) to get rid of the matrix multiplication; ($b$) $A\hat{\mathbf{g}}$ can be computed in $O(|\mathcal{E}|)$ with sparse matrix multiplication; ($c$) $D \_+^{-1}$ is an $O(N)$ operation since it is scaling over all elements in the vector. In all, the time/memory complexity of GRASP is $O(N+|\mathcal{E}|+n)$.

---

> ### Author Response · Authors · 2023-11-18
> **Response - part 2**
>
> (Connecting to the Q2 above ^ )
> - **Empirical results.**
> Then we conduct comprehensive experiments comparing the time and space costs of our algorithm with various baselines (both post-hoc and training-based approaches). From the experimental results, it is evident that our algorithm is highly efficient across all datasets.
>     - We first present the time(s)/memory(M) of all post-hoc baseline methods consumed over all datasets considered.
>
>         |            | cora       |           | amazon    |           | coauthor-cs |         | chameleon |         |
>         |:-----------|:-----------|:----------|:----------|:----------|:------------|:--------|:----------|:--------|
>         |  | time       | memory    | time      | memory    | time        | memory  | time      | memory  |
>         | MSP        | 0.03       | 622.79    | 0.04      | 635.41    | 0.13        | 1092.05 | 0.05      | 634.05  |
>         | Energy     | 0.05       | 624.10    | 0.10      | 636.52    | 0.11        | 1094.72 | 0.11      | 634.75  |
>         | KNN        | 0.24       | 630.92    | 0.13      | 658.21    | 0.30        | 1101.30 | 0.17      | 636.44  |
>         | ODIN       | 0.02       | 621.82    | 0.05      | 636.40    | 0.07        | 1093.64 | 0.16      | 635.41  |
>         | Mahalanobis| 0.15       | 627.39    | 0.22      | 641.21    | 0.29        | 1099.38 | 0.21      | 637.18  |
>         | GNNSafe    | 0.06       | 626.23    | 0.15      | 652.80    | 0.35        | 1106.19 | 0.12      | 637.95  |
>         | **GRASP**  | **0.05**       | **628.79**    | **0.15**      | **677.26**    | **0.25**        | **1118.02** | **0.15**      | **644.09**  |
>
>
>         |            | squirrel   |           | arxiv-year |           | snap-patents |         | wiki     |         |
>         |:-----------|:-----------|:----------|:-----------|:----------|:-------------|:--------|:---------|:--------|
>         |   | time       | memory    | time       | memory    | time         | memory  | time     | memory  |
>         | MSP        | 0.05       | 658.00    | 0.36       | 726.59    | 3.41         | 4014.22 | 3.01     | 9765.80 |
>         | Energy     | 0.12       | 660.50    | 0.52       | 726.17    | 3.24         | 4039.91 | 3.13     | 9788.45 |
>         | KNN        | 0.14       | 660.18    | 3.77       | 732.90    | 6241.01      | 3933.32 | 273.96   | 9812.89 |
>         | ODIN       | 0.06       | 659.58    | 0.61       | 727.68    | 3.61         | 4005.23 | 3.00     | 9760.24 |
>         | Mahalanobis| 0.19       | 661.44    | 1.28       | 730.41    | 5.93         | 4056.08 | 6.17     | 9829.16 |
>         | GNNSafe    | 0.10       | 662.48    | 0.97       | 790.20    | 13.48        | 4187.67 | 62.09    | 9655.11 |
>         | **GRASP**  | **0.13**       | **668.07**    | **0.68**       | **768.21**    | **13.63**        | **4279.92** | **218.67**   | **9824.79** |
>
>     - Additionally, we provide comparison with training-based methods in the literature. It is worth noting that the training-based methods incur substantial time and space costs due to training from scratch,  exhibit a significant performance gap compared to our algorithm and therefore are practically infeasible for practical deployment.
>
>         |			|cora		|			|amazon		|			|coauthor-cs|	      |chameleon|         |
>         |:--------  |:--------|:--------|:--------|:--------|:--------|:--------|:--------|:--------|
>         ||time		|memory		|	time	|memory		|time		|memory   |time		|memory   |
>         | GKDE [1]  | 26.72  | 3366.01   | 160.00   | 3388.59   | 1004.73   | 4199.84  | 37.24   | 3310.92  |
>         | GPN [2]  | 28.64  | 3702.57   | 58.72    | 3719.10   | 77.55     | 3634.66  | 50.65   | 3620.97  |
>         | OODGAT [3] | 77.99  | 3369.00   | 395.88   | 3390.88   | 411.44    | 3851.37  | 182.11  | 3291.63  |
>         | **GRASP**   | **0.05**   | **628.79**    | **0.15**     | **677.26**    | **0.25**      | **1118.02**  | **0.15**    | **644.09**  |
>
>
>         |            | squirrel   |           | arxiv-year |           | snap-patents |         | wiki     |         |
>         |:-----------|:-----------|:----------|:-----------|:----------|:-------------|:--------|:---------|:--------|
>         ||time		|memory		|	time	|memory		|time		|memory   |time		|memory   |
>         | GKDE  [1]    | 170.26     | 3234.38   | OOT        | -         | -            | OOM     | -        | OOM     |
>         | GPN [2]   | 50.70      | 3650.58   | 444.84     | 3808.72   | -            | OOM     | -        | OOM     |
>         | OODGAT  [3]  | 454.62     | 3319.99   | 1214.98    | 3491.11   | -            | OOM     | -        | OOM     |
>         | **GRASP**  | **0.13**       | **668.07**    | **0.68**       | **768.21**    | **13.63**        | **4279.92** | **218.67**  | **9824.79** |

---

> ### Author Response · Authors · 2023-11-18
> **Response - part 3**
>
> (Connecting to the Q2 above ^ )
>
> Based on the  above experimental results, we have:
>
> - In comparison to training-based methods, post-hoc methods exhibit significantly lower runtime and memory consumption.
> - Considering that the minimum time and space complexity required to run a graph algorithm is $O(N+|\mathcal{E}|)$, as outlined in our algorithm complexity analysis, our algorithm incurs limited additional time and space costs, specifically $O(n)$. This can be validated by the small extra overhead incurred by our algorithm compared to MSP or Energy (8 times n) from the table.
> - Compared with training-based methods, our algorithm demonstrates substantially lower time and space demands on three large-scale datasets. And when compared to post-hoc baselines on these datasets, the overhead of our method is also reasonable. The performance of our method on these three large-scale datasets underscores the strong practicality of our approach.
>
>
> [1] Xujiang Zhao, Feng Chen, Shu Hu and Jin-Hee Cho. Uncertainty Aware Semi-Supervised Learning on Graph Data. NeurIPS 2020.
>
> [2] Maximilian Stadler, Bertrand Charpentier, Simon Geisler, Daniel Zügner and Stephan Günnemann. Graph Posterior Network: Bayesian Predictive Uncertainty for Node Classification. NeurIPS 2021.
>
> [3] Yu Song and Donglin Wang. Learning on Graphs with Out-of-Distribution Nodes. KDD 2022.
>
>
> > **Q3. "Why does the theoretical analysis assume that the edges follow a Bernoulli distribution?"**
>
> As we are dealing with discrete graphs, where edges either exist or do not, the adjacency matrix values are binary, taking on either 0 or 1. This naturally aligns with the Bernoulli distribution, which is frequently employed in graph structure learning, as exemplified in several pertinent references in [1,2,3].
>
> [1] Luca Franceschi, Mathias Niepert, Massimiliano Pontil and Xiao He. Learning Discrete Structures for Graph Neural Networks. ICML 2019.
>
> [2] Pantelis Elinas, Edwin V. Bonilla and Louis C. Tiao. Variational Inference for Graph Convolutional Networks in the Absence of Graph Data and Adversarial Settings. NeurIPS 2020.
>
> [3] Tong Zhao, Yozen Liu, Leonardo Neves, Oliver Woodford, Meng Jiang and Neil Shah. Data Augmentation for Graph Neural Networks. AAAI 2021.
>
>
> > **Q4. "The impact of other propagation mechanisms, like higher-order diffusion."**
>
> Excellent suggestions! We have conducted an analysis of the hyper-parameter $k$ (order of propagation) and its impact on the AUROC, averaged across all datasets using a GCN backbone. Our findings indicate that OOD detection performance benefits from propagation orders within a reasonable range. However, excessive propagation (greater than 10) may be detrimental.
>
>
> | |  |  |  |  | |
> |:-------- |:--------:|:--------:|:--------:|:--------:| :--------:|
> |$k$|2|4|8|10|16|
> |AUROC|79.38 |80.87 |82.45|80.53 |71.19 |
>
>
>
> In addition, we have investigated various classical propagation mechanisms, including Personalized PageRank (PPR) [1], Heat Kernel Diffusion (GraphHeat) [2], Graph Diffusion Convolution (GDC) [3], Mixing Higher-Order Propagation (MixHop) [4], and Generalized PageRank (GPR) [5]. The results (**AUROC** in percentages) demonstrate that GRASP emerges as the most effective propagation strategy in our study.
>
> | |cora| amazon| coauthor| chameleon|
> |:-------- |:--------:|:--------:|:--------:|:--------:|
> |PPR|82.68 |83.34 |85.18 |52.79 |
> |GraphHeat	|92.76 	|76.99 |	70.53 |	58.48 |
> |GDC|89.48 |92.33 |96.52 |54.47 |
> |MixHop|79.80 |82.68 |86.04 |55.03 |
> |GPR|78.70 |81.57 |81.84 |52.86 |
> |**GRASP**|**94.65** |**96.76** |**97.94** |**67.97** |
>
>
> | |squirrel| arxiv-year| snap-patents| wiki |
> |:-------- |:--------:|:--------:|:--------:|:--------:|
> |PPR|48.17 |    56.87 |46.70 |34.83 |
> |GraphHeat	|45.63 |38.40 |	41.00 |OOM |
> |GDC|48.82  | OOM|OOM|OOM  |
> |MixHop|48.66 | 34.41 |28.93 |39.22 |
> |GPR|49.64 | 34.30 |29.66 |36.98 |
> |**GRASP**|**54.93** | **74.66** |**67.36** |**65.56** |
>
>
>
> [1] Johannes Gasteiger, Aleksandar Bojchevski, and Stephan Günnemann. Predict then Propagate: Graph Neural Networks Meet Personalized PageRank. ICLR 2019.
>
> [2] Bingbing Xu , Huawei Shen , Qi Cao , Keting Cen and Xueqi Cheng. Graph Convolutional Networks using Heat Kernel for Semi-supervised Learning. IJCAI 2019.
>
> [3] Johannes Gasteiger, Stefan Weißenberger and Stephan Günnemann. Diffusion Improves Graph Learning. NeurIPS 2019.
>
> [4] Sami Abu-El-Haija, Bryan Perozzi, Amol Kapoor, Nazanin Alipourfard, Kristina Lerman, Hrayr Harutyunyan, Greg Ver Steeg, and Aram Galstyan. MixHop: Higher-Order Graph Convolutional Architectures via Sparsified Neighborhood Mixing. ICML 2019.
>
> [5] Eli Chien, Jianhao Peng, Pan Li, and Olgica Milenkovic. Adaptive universal generalized pagerank graph neural network. ICLR 2021.
>
>
> Finally, we would like to express our gratitude for the insightful questions raised by the reviewer. We would be more than grateful if the reviewer would like to champion our paper in the discussion phase. Thanks!

---

### Official Review · Reviewer_EGVF · 2023-10-31

**Soundness:** 2 fair
**Presentation:** 3 good
**Contribution:** 2 fair
**Rating:** 5
**Confidence:** 5

**Summary:**

The paper studies graph OOD detection through OOD score propagation. The paper theoretically proves that propagation can enhance OOD detection when there are more intra-edges within ID/OOD nodes than inter-edges between ID and OOD nodes. The paper further proves that the efficacy of propagation can be improved by adding edges to nodes that have more connections to ID nodes than to OOD nodes. Then, the paper designs a simple augmentation strategy to boost the performance of propagation.  Experimental results demonstrate the effectiveness of their proposed strategy.

**Strengths:**

- The paper theoretically analyzed when OOD score propagation will work and how to boost its performance.
- The proposed method is simple and experimental results validate its effectiveness.
- This paper is well-written and easy to understand.

**Weaknesses:**

- The proposed method is developed based the assumption that intra-edges dominate the graph. However, the challenge in OOD detection arises from the unknown pattern of OOD nodes. This strong assumption makes the analysis less insightful and constrains the practicality of the proposed method.
- The experiments are conducted exclusively with one type of OOD nodes (Label Leave-Out), which limits the comprehensive evaluation of the proposed method under different distribution shifts.
- The selection of Sid/Sood is based on the pre-computed OOD scores (using MSP), rendering the proposed method ineffective when MSP fails, as observed in dataset Squirrel in Table 2.

**Questions:**

- According to Theorem 3.2, the propagation is deemed effective only when intra-edges dominate. Why does it seem to work well in heterophily dataset Chameleon?
- Why does the improvement of the proposed method over GNNSafe appear to be marginal in homophily datasets like Amazon and Coauthor, while the improvement is more significant in the two heterophily datasets in Table 2?
- How does the proposed method perform when faced with different types of OOD, such as structural manipulation and feature interpolation as addressed in GNNSafe?
- Is the proposed method sensitive to hyperparameter k, $\alpha$, and $\beta$? How to choose the $\alpha$ and $\beta$ when MSP’s performance varies in different datasets?

---

> ### Author Response · Authors · 2023-11-18
> **Response - Part 1**
>
> We thank the reviewer for the insightful questions! Below we address each of your comments in detail.
>
> > **1. "Assumption of the intra-edge domination."**
>
> We respectfully point out that the reviewer's interpretation of "*The proposed method is developed based on the assumption that intra-edges dominate the graph*" is incorrect. The intra-edge-domination assumption is only required for the *naive propagation method* as proved in Theorem 3.2. One of the primary objectives of our algorithm is to address the hard problem of graph node ood detection in heterophilic scenarios, **where intra-edges do not dominate the graph.**
>
> For instance, on three particularly challenging heterophily datasets—Chameleon, arXiv-year, and snap-patents, the application of naive propagation method leads to a significant performance decline. In contrast, our approach not only reversed this trend but also demonstrates substantial improvement compared to pre-propagation performance. Refer to the table below for detailed results, where the performance values  are evaluated on backbone GCN. From the table we can see that our method yields a maxmium performance increase of up to **32.5** percentage points on these challenging heterophilic datasets (AUROC on arXiv-year for MSP). This robustly underscores the effectiveness of our proposed approach.
>
> | | Chameleon| |arXiv-year| |snap-patents ||
> |:-------- |:--------| :--------|:--------|:--------|:--------|:--------|
> ||AUROC	|FPR95	|AUROC		|FPR95		|AUROC		|FPR95	|
> |MSP|59.91|90.87| 43.35|95.60 | 51.42|93.10|
> |MSP+prop|42.82|98.17 |  35.30|100.0  | 27.25|100.0  |
> |MSP+**GRASP**|**69.57**|**72.77** | **75.87**|**85.78** | **67.80**|**74.91** |
> |Energy	|59.68|94.98|47.58|95.00 |46.93|93.85 |
> |Energy+prop|49.65|97.74 | 35.40|100.0   |  27.26|100.0 |
> |Energy+**GRASP**|**67.97**|**74.54**| **74.66**|**86.70** | **67.36**|**74.60** |
>
>
> > **2. "How does the proposed method perform when faced with structural manipulation and feature interpolation as addressed in GNNSafe?"**
>
> Thanks for raising this point! We adopt the "label leave-out" setting since it aligns closely with practical scenarios where OOD data comes from a different class. The other two settings (structural manipulation and feature interpolation) construct OOD data through manual perturbation, which significantly differs from real-world OOD datas. Even if a method performs well on artificially synthesized OOD data, it does not necessarily reflect its real performance in real-world scenarios. Therefore, we exclusively adopt the "label leave-out" setting.
>
>
> As the reviewer suggest, we also conduct experiments on the two different types of OOD (structural manipulation and feature interpolation) proposed by GNNSafe, and the results (AUROC) indicate that our method still performs well in these different constructed OOD scenarios.
>
> |			|cora	|		|Amazon	|		|coauthor|		|Average|	  |
> |:-------- |:--------| :--------|:--------|:--------|:--------|:--------|:--------|:--------|
> ||S		|F 		|S		|F 		|S		|F 		|S		|F    |
> |MSP			|70.90 	|85.39 	|98.27 	|97.31 	|95.30 	|97.05 	|88.16 	|93.25|
> |ODIN		|49.92 	|49.88 	|93.24 	|81.15 	|52.14 	|51.54 	|65.10 	|60.86|
> |Mahalanobis	|46.68 	|49.93 	|71.69 	|76.50 	|80.46 	|93.23 	|66.28 	|73.22|
> |Energy		|71.73 	|86.15 	|98.51 	|97.87 	|96.18 	|97.88 	|88.81 	|93.97|
> |GKDE		|68.61 	|82.79 	|76.39 	|58.96 	|65.87 	|80.69 	|70.29 	|74.15|
> |GPN			|77.47 	|85.88 	|97.17 	|87.91 	|34.67 	|72.56 	|69.77 	|82.12|
> |GNNSafe		|87.52 	|93.44 	|99.58 	|98.55 	|99.60 	|99.64 	|95.57 	|97.21|
> |**GRASP**		|**95.29** 	|**98.93** 	|**98.50** 	|**98.45** 	|**99.99** 	|**99.80** 	|**97.93** 	|**99.06**|

---

> ### Author Response · Authors · 2023-11-18
> **Response - Part 2**
>
> > **3. "The proposed method may be ineffective when MSP fails, as observed in dataset Squirrel in Table 2."**
>
> We would like to share a differing viewpoint from the reviewer.  The effectiveness of our method, GRASP, is evidenced through two key observations:
>
> 1. As shown in Table 2, GRASP significantly surpasses MSP by a margin of **6.76** in AUROC on the Squirrel dataset using the GCN model. This improvement underscores the benefits of score propagation augmented by our method. **Notably, this enhancement is also observed in scenarios where MSP alone is ineffective**.
>
> 2. On the other hand, the efficacy of our strategy is primarily measured by the extent to which it enhances Out-of-Distribution (OOD) detection performance beyond the initial OOD scores prior to propagation. Among traditional OOD detection methods, the KNN scores achieved the highest AUROC on the Squirrel dataset. However, when our propagation technique is applied to KNN, there is a notable improvement in its pre-propagation performance, as demonstrated in the table below. Furthermore, across all datasets, applying our method to KNN consistently improves performance in terms of both AUROC and FPR95. This consistently better performance across various metrics highlights the robustness of our approach. It is important to note that this improvement is observed even in cases where MSP underperforms.
>
>     ||cora	|		|amazon		|			|coauthor	|		|chameleon	|		|squirrel	|      |
>     |:--------|:--------|:--------|:--------|:--------|:--------|:--------|:--------|:--------|:--------|:--------|
>     ||AUROC	|FPR95	|AUROC		|FPR95		|AUROC		|FPR95	|AUROC		|FPR95	|AUROC |FPR95 |
>     |KNN		|81.24	|72.29	|86.01		|60.61		|91.34		|47.99	|62.09		|93.43	|56.74	|94.42 |
>     |KNN+**GRASP**|**88.23**|**33.16**|**95.30**|**21.43**|**97.57**|**8.86** |**74.44**		|**62.29**	|**58.57**		|**87.96** |
>
>
> We are also more than happy to provide more discussion to address the concerns regarding the "MSP reliance" as follows:
> 1.  As proved in Theorem 4.2, the conditions required for our method to work well are easy to satisfy. The chosen node set G only needs to satisfy the condition $|\mathcal{E} \_{G\leftrightarrow S \_{ID}}|>|\mathcal{E} \_{G \leftrightarrow S \_{OOD}}|$.
> 2.  Our method selectively considers data points at the two ends of the score distribution. This selection minimizes the error in selecting $S \_{ID}$ and $S \_{OOD}$.
>
>
> > **4. "Why naive propagration seems to work well in heterophily dataset Chameleon?"**
>
> We are happy to provide more clarification here.  As illustrated in Table 3, on the Chameleon dataset, both MSP and Energy exhibit **degraded** performance after naive propagation (evaluated using FPR95, where lower values indicate better results). Notably, our proposed method successfully reverses this trend, significantly improving performance. Table 3 provides a clear demonstration of this enhancement.
>
>
>
> > **5. "Why does the improvement of the proposed method over GNNSafe appear to be marginal in homophily datasets, while the improvement is more significant in the heterophily datasets?"**
>
> We respectfully disagree with the reviewer's observation. Firstly, our method's improvement over GNNSafe in homophily datasets is not marginal, which can be verified by the results in the following table (Averaged on the three homophily datasets Cora, Amazon and Coauthor). From the results we can see that our method outperforms GNNSafe by **7.10** and **1.62** on FPR and AUROC respectively. Secondly, the more pronounced improvement on heterophily datasets aligns with our theoretical derivation, since the augmentation strategy aims to improve the ratio of "intra-edges" after propagation. Notably, if inter-edges dominate, traditional propagation methods would lead to performance degradation, while our approach successfully reverses this trend, as evidenced by certain results in Table 3.
> |		|FPR95|AUROC|
> |:--------|:--------|:--------|
> |GNNSafe|22.25|94.83	|
> |**GRASP**|**15.15**|**96.45**	|
>
>
> > **6. "Hyperparameter sensitive analysis and how to choose $\alpha$ and $\beta$ when MSP’s performance varies in different datasets"**
>
> Great suggestions! We conduct a sensitivity analysis of all the hyper-parameters on all the datasets and the averaged results (**AUROC** in percentages) on model GCN with Energy are displayed in the following table. The performance comparison in the table for each hyper-parameter is reported by fixing other hyper-parameters. From the results we can see that within the range of chosen hyperparameter values, our proposed method's performance does not vary significantly, which constantly outperforms baselines by a large margin.
>
>
> | |  |  |  |  | |
> |:-------- |:--------:|:--------:|:--------:|:--------:| :--------:|
> | $\alpha$ | 1 | 2.5 | 5 | 10 |15 |
> |AUROC |81.23 |81.24 |82.45|80.83 |80.35 |
> |$\beta$|10|40|50|60|90|
> |AUROC|77.78 |81.22 |82.45|81.79  |80.12 |
> |$k$|2|4|8|10|16|
> |AUROC|79.38 |80.87 |82.45|80.53 |71.19 |

---

> ### Author Response · Authors · 2023-11-21
> **Follow-up on the response**
>
> Dear Reviewer,
>
> We are eager to receive your feedback and any suggestions you might have. The insights from you would be invaluable in enhancing the quality of this work. If there is any additional information or clarification needed from our end, please feel free to let us know.
>
> Thank you once again for your time and consideration. We look forward to hearing from you soon.
>
> Best,
>
> Authors

---

> ### Comment · Reviewer_EGVF · 2023-11-23
> **Concerns**
>
> Thanks for the response that addresses part of my concerns. In your paper, Theorem 3.2 is as important as a theorem presented before technical details, and it is important to motivate and justify your method. However, after I raised my concern, in your response, you are downplaying the importance of it. This confuses me a lot. Also, on squirrel data, the question of why your method is not good is not well-justified for the heavy dependency on MSP to precompute OOD scores. Although you provided results on different types of OODs, and the results show that your method is better, it remains unclear how this experiment is done, and it is unclear if the results are reproducible, since no code is released to reviewers.

---

> ### Author Response · Authors · 2023-11-23
> **Response to R2's concerns**
>
> Thank you for your valuable feedback! We are pleased that our response has addressed some of your concerns, and we are more than willing to address the additional issues you have.
>
> > In your paper, Theorem 3.2 is as important as a theorem presented before technical details, and it is important to motivate and justify your method. However, after I raised my concern, in your response, you are downplaying the importance of it
>
> We respectfully disagree with the reviewer's perspective. Theorem 3.2 indicates that only when intra-edges dominate, direct propagation on $A$ (naive propagation) can enhance OOD detection performance. However, if intra-edges do not dominate, naive propagation does not guarantee performance improvement. The significance of Theorem 3.2 lies in highlighting the limitation of naive propagation, and our method is designed to address this limitation. The experimental results provided in our previous responses serve as empirical evidence supporting these points. Therefore, we do not agree with the reviewer's statement regarding "downplaying the importance of Theorem 3.2." We hope this clarification addresses the reviewer's concerns.
>
> > Also, on squirrel data, the question of why your method is not good is not well-justified for the heavy dependency on MSP to precompute OOD scores.
>
> Regarding this point, we would like to provide further clarifications:
> The results of our proposed method presented in Table 2  (denoted as **GRASP (ours)** in the "OOD Detection Method" column) are the performance after propagating on the augmented $A$, using the pre-propagated Energy score. Below, we present the performance of MSP, Energy, and KNN before and after using our method respectively. As shown in the table, there is a significant improvement in performance for all datasets when various scores are propagated using our method. Notably, on the Squirrel dataset, the performance of MSP, after using our method, surpasses that of KNN, even when MSP itself is ineffective before propagation (MSP's AUROC is only 48.17 on Squirrel before propagation). This demonstrates the superiority of our method over all baselines.
>
> ||cora	|		|amazon		|			|coauthor	|		|chameleon	|		|squirrel	|      |
> |:--------|:--------|:--------|:--------|:--------|:--------|:--------|:--------|:--------|:--------|:--------|
> |method		|AUROC	|FPR95	|AUROC	|FPR95	|AUROC		|FPR95	|AUROC		|FPR95	|AUROC		|FPR95|
> |MSP			|89.33 	|52.23 	|90.47 	|49.52 	|95.29 		|23.87 	|59.91 		|90.87 	|48.17 		|91.99|
> |MSP+prop	|91.78 	|31.24 	|95.37 	|26.87 	|97.70 		|8.74 	|42.82 		|98.17 	|53.26 		|91.77|
> |MSP+**GRASP**	|**95.13** 	|**19.31** 	|**96.86** 	|**13.58** 	|**97.92** 		|**7.80**	|**69.57** 		|**72.77** 	|**57.99** 		|**88.10**|
> |Energy		|89.48 	|52.05 	|92.33 	|39.49 	|96.52 		|14.98 	|59.68 		|94.98 	|45.06 		|94.29|
> |Energy+prop	|90.34 	|38.03 	|96.36 	|23.42 	|96.89 		|11.86 	|49.65 		|97.77 	|53.95 		|91.00|
> |Energy+**GRASP**|**94.65** 	|**21.92** 	|**96.76** 	|**15.64** 	|**97.94** 		|**7.88** 	|**67.97** 		|**74.54** 	|**54.93** 		|**90.21**|
> |KNN			|81.24 	|72.29 	|86.01 	|60.61 	|91.34 		|47.99 	|62.09 		|93.43 	|56.74 		|94.42|
> |KNN+prop	|84.10 	|59.43 	|93.01 	|38.95 	|96.19 		|15.05 	|61.63 		|92.71 	|58.43 		|92.48|
> |KNN+**GRASP**	|**88.23** 	|**38.49** 	|**95.30** 	|**22.59** 	|**97.57** 		|**8.72** 	|**74.44** 		|**64.61** 	|**58.57** 		|**88.59**|
>
> We hope our explanations  can alleviate the reviewer's concerns.
>
> Thank you for your attention!

---

### Official Review · Reviewer_hJQH · 2023-11-01

**Soundness:** 2 fair
**Presentation:** 3 good
**Contribution:** 3 good
**Rating:** 6
**Confidence:** 3

**Summary:**

This paper study detecting Out-of-Distribution (OOD) nodes on graphs. The authors first demonstrate through empirical and theoretical evidence that previous OOD detection methods relying on information propagation are only applicable to scenarios where the number of intra-edges is greater than inter-edges. Following this, based on their analytical conclusions, the authors propose an edge augmentation strategy called GRASP to enhance the effectiveness of these methods. Experimental results on several datasets demonstrate the effectiveness of their approach.

**Strengths:**

A. This paper is well organized and written, with a clear definition of the research problem and detailed introductions of the motivation and methodology. Key conclusions are clearly marked in the paper.

B. The proposed method has good generalizability. The post-processing strategy does not require retraining the model, allowing for flexible application to various existing methods and improving their effectiveness on OOD node detection task.

C. The experimental results are impressive. The authors have compared their method with baselines on many datasets and conducted a detailed analysis of the experimental results, showing that the proposed method can significantly enhance OOD node detection.

**Weaknesses:**

A. The data augmentation relies on a robust base model. During the graph augmentation, ID nodes and OOD nodes are sampled based on the base model's OOD prediction scores, which may lead to potential error propagation.

B. Analysis of some key hyper-parameters is missing. It appears that the two hyper-parameters \alpha and \beta in the proposed method significantly affect the sampling results, and I would like to know the impact of different \alpha and \beta values on the results.

**Questions:**

A. According to the authors, the number of intra-edges and inter-edges has a significant impact on OOD detection. What are the respective proportions of these two types of edges in different datasets? And what are their proportions after graph augmentation?

---

> ### Author Response · Authors · 2023-11-18
> **Response**
>
> We thank the reviewer for the insightful questions! Below we address each of your comments in detail.
>
> > **Q1. "The base model's OOD prediction scores may lead to potential error propagation"**
>
> We are more than happy to provide more discussion to address the concerns as follows:
>
> 1. We would like to highlight that our augmentation approach will not incorporate error-edges connecting ID and OOD, since we add edges in the training dataset and the augmentation is performed **exclusively** on in-distribution (ID) data.
> 2. As proved in Theorem 4.2, the conditions required for our method to work well are easy to satisfy. The chosen node set G only needs to satisfy the condition $|\mathcal{E} \_{G\leftrightarrow S \_{ID}}|>|\mathcal{E} \_{G \leftrightarrow S \_{OOD}}|$. This condition, compared to directly augmenting edges on the test set, allows for a stronger degree of tolerance to errors.
> 3. Our method selectively considers data points at the two ends of the Maximum Softmax Probability (MSP) distribution. This selection minimizes the error in selecting $S \_{ID}$ and $S \_{OOD}$.
>
> For reviewer's interest, we are happy to provide comparison with an alternative approach -- augmenting edges directly on the test data. Such strategy would lead to severe error propagation, which is known as "confirmation bias". To validate this point, we conduct experiments comparing the effectiveness of "directly augmenting edges on test data" versus "our proposed method of augmenting edges in the training dataset." From the results, we can see that our method has achieved an improvement of nearly **10** percentage points in AUROC compared to the approach of directly adding edges on test data, which supports our assertion.
>
> (Cells' values represent AUROC in percentages on backbone GCN with Energy Score.)
>
> | Method| Cora | Amazon | Coauthor | Chameleon | Squirrel | Average |
> |:-------- |:--------:|:--------:|:--------:|:--------:| :--------:| :--------:|
> |Before Augmetation (Energy Score)|89.48|92.33|96.52|59.68|45.06|76.61|
> |Augmentation on Test (with confirmation bias)|85.90|87.99|94.72|48.32|46.96|72.78|
> |Augmentation on Train (**Ours**)|**94.65**|**96.76**|**97.94**|**67.97**|**54.93**|**82.45**|
>
> > **Q2. "Analysis of two hyper-parameters $\alpha$ and $\beta$ is missing"**
>
> Great suggestions! We conduct sensitivity analysis of hyper-parameters $\alpha$ and $\beta$ on all the datasets and the averaged results (**AUROC** in percentages) on model GCN with Energy are displayed in the following table. The performance comparison in the table for each hyper-parameter is reported by fixing other hyper-parameters. From the results we can see that within the range of chosen hyperparameter values, our proposed method's performance does not vary significantly, which constantly outperforms baselines by a large margin.
>
>
>
> |          |       |       |       |       |       |
> |:-------- |:-----:|:-----:|:-----:|:-----:| -----:|
> | $\alpha$ |   1   |  2.5  |   5   |  10   |    15 |
> | AUROC    | 81.23 | 81.24 | 82.45 | 80.83 | 80.35 |
> | $\beta$  |  10   |  40   |  50   |  60   |    90 |
> | AUROC    | 77.78 | 81.22 | 82.45 | 81.79 | 80.12 |
>
> > **Q3. "The respective proportions of intra-edges and inter-edges before and after graph augmentation"**
>
> Great suggestions! We provide proportions of intra-edges and inter-edges after two rounds of propagations in each dataset  before and after graph augmentation respectively in the following table. From the results we can see that the proportions of intra-edges increase after graph augmentation, which lead to improved ood performance, validating the correctness of our theory.
>
> |Dataset| intra-edges ratio(Before Aug) | inter-edges ratio(Before Aug) | intra-edges ratio(After Aug) | inter-edges ratio(After Aug) |
> |:--------|:--------:|:--------:|:--------:|:--------:|
> |cora		|0.8155 	|0.1845 	|**0.9188** 	|0.0812|
> |amazon-photo|0.6405 	|0.3595 	|**0.7264** 	|0.2736|
> |coauthor-cs	|0.8105 	|0.1895 	|**0.8690** 	|0.1310|
> |chameleon	|0.5082 	|0.4918 	|**0.5274** 	|0.4726|
> |squirrel	|0.4972 	|0.5028 	|**0.5167** 	|0.4833|
>
> Finally, we would like to express our gratitude for the insightful questions raised by the reviewer. We would be more than grateful if the reviewer would like to champion our paper in the discussion phase. Thanks!

---

### Author Response · Authors · 2023-11-18
**Summary of response – thanks to all reviewers for the thorough and insightful feedback**

We thank all the reviewers for their constructive and valuable feedback. As abbreviations, we refer to Reviewer hJQH as R1, Reviewer EGVF as R2, and Reviewer 13zH as R3 respectively.

We are honored that **all** the reviewers acknowledge the effectiveness of our proposed method (R1, R2, R3) with generalizable (R1), simple (R2) and original augmentation strategy (R3).
Multiple reviewers value the theoretical nature of our paper (R2, R3), finding theoretical analysis to be rigorous (R3) and clearly proved (R3). Beyond theoretical insight, the reviewers recognized the impressive (R1) and effective (R2) experimental results. We are glad that **all** the reviewers comment that our paper is well-written (R1, R2, R3), easy to understand (R2) and has clear motivation, definition and structure (R1, R3) of the research problem. We also appreciate R1's acknowledgment of the flexibility, good generalizability and practical applicability of our method.

We have addressed the reviewers’ comments and concerns in individual responses to each reviewer.

---

### Meta-Review · Area_Chair_VZVb · 2023-12-07

**Metareview:**

This work considers the task of detecting Out-of-Distribution (OOD) nodes on graphs. The key insight reminds me a bit of collective classification (https://en.wikipedia.org/wiki/Collective_classification), which does score propagation on a graph (initially proposed by (Wu et al., 2022)). Wu et al. talks about label propagation in their paper, while this paper just cites Wu, Ghahramani, and Lafferty without explaining the connection to collective inference. Doing score propagation to help OOD detection is interesting. One key idea in Section 4 is to randomly connect nodes with known in-distribution nodes and check how the propagation procedure is different: This is used just as a way to better score the OOD-ness of a node after propagation.

The key assumption is that a notion of homophily between IND and OOD nodes (as clearly stated in the theorems). That is very simplistic but I am OK with that if there are strong theoretical results.

The paper has some interesting ideas (reviewers also indicate so). I hoped to find more substantial theoretical contributions (proving more complex properties based on IND and OOD nodes... for instance, collective inference also talks about how to do a theoretically-sound multiple-round propagation, where the topological properties of how IND and OOD nodes are connected would be relevant [e.g., how clustered they are]]). How does multiple propagation rounds depend on the topology (rather than 1-hop connectivity) of how in- and out- nodes are connected in the graph? The authors try to get around this complexity with the heuristic in Section 4. Since the edges are added at random to known IND nodes, in average there is a deterministic mapping between the score in Theorem 3.2 and Theorem 4.2 for any given graph. So, there is no real improvement in the score, it is just a Monte Carlo way to scale and somewhat take care of the topology of IND and OOD connectivity without having to perform an actual calculation. That is fine. But it would be nice to see a proof of how this approach will overcome the topology issues in the propagation (say, a stochastic block model with a spectral gap result would be interesting; or maybe a worst-case analysis w.r.t. the topology of IND and OOD connections).

Reading the paper, reviews, and rebuttal, I could not find a precise definition of "OOD", but the paper seems to assume the representation (or the downstream preditor) will capture the "OOD-ness" of a set of nodes through some known property (e.g., low/high confidence in predictions). This is a common assumption in OOD detection for vision tasks, which is now permeating to the graph OOD literature and is the same assumption (Wu et al., 2022) used. It is important to emphasize this is just an assumption not a provable property of OOD-ness of nodes.

More importantly, since none of the reviewers want to champion the paper in our private discussion, my recommendation is at the lower end of borderline. The paper has interesting ideas and could be substantially improved if the authors are interested in strengthening the theoretical results.

**Justification For Why Not Higher Score:**

While I think the paper has good ideas, it needs to mature more. Not using multiple rounds of OOD-ness propagation makes the theory too simple and the method not as powerful as it could be.

**Justification For Why Not Lower Score:**

N/A

---

### Decision · Program_Chairs · 2024-01-16

Reject